# Stability Analysis and Augmentation Design of a Bionic Multi-Section Variable-Sweep-Wing UAV Based on the Centroid Self-Trim Compensation Morphing

Hang Ma ⬨, Yuxue Ge *, Bifeng Song and Yang Pei

School of Aeronautics, Northwestern Polytechnical University, Xi'an 710072, China; davidma@mail.nwpu.edu.cn (H.M.); sbf@nwpu.edu.cn (B.S.); peiyang_yang@nwpu.edu.cn (Y.P.)
* Correspondence: ge_yuxue@nwpu.edu.cn

**Abstract:** In this study, a design scheme for a high-aspect-ratio bionic multi-section variable-sweep wing unmanned aerial vehicle (UAV) that utilizes the reverse coordinated change in the sweep angle of the inner and outer wing sections is proposed, which improves the aerodynamic performance and realizes the self-trim compensation of the wing's centroid. According to the layout characteristics of this type of UAV, a reasonable distribution design of the wingspan ratio of the inner and outer sections is explored, to reduce the impact of aerodynamic center movement and moment of inertia change. The calculation and analysis results show that the coordinated variable-sweep scheme can significantly improve the influence of sweep angle change on the longitudinal static stability margin of UAVs with a high aspect ratio. The coordinated sweep angle change in the inner and outer wing sections can not only reduce the drag during high-speed flight, but also play a significant role in improving the performance of the aircraft at different stages in the mission profile. Appropriately increasing the wingspan proportion of the inner section can reduce the trim resistance of the V-tail, reduce the thrust of the engine, and increase the range and duration of the UAV. From the perspective of stability change, the multi-section variable-sweep wing UAV with a wingspan ratio of the inner and outer sections that is between 1.41 and 1.78 has better dynamic stability performance. Among them, the UAV with a wingspan ratio of the inner and outer sections that is equal to 1.41 has better longitudinal stability performance, while the UAV with a wingspan ratio of the inner and outer sections that is equal to 1.78 has better lateral/directional stability performance.

**Keywords:** bionic multi-section variable-sweep wing unmanned aerial vehicle; centroid self-trim compensation morphing; aerodynamic analysis; stability analysis

## 1. Introduction

Morphing aircraft can improve the aerodynamic performance, flexibility of mission execution, and comprehensive performance within the flight envelope, by changing the local or overall shape [1–5]. It is found that the performance improvement caused by wing deformation increases significantly with the size and deformation scale of the aircraft. However, the aerodynamic and dynamic characteristics of a high-aspect-ratio wing aircraft change drastically during the morphing process. Changes in parameters such as the wingspan, wing area, and moment of inertia will cause the centroid and aerodynamic center to move in a large range, which may affect the stability, maneuverability, and flight performance, such as endurance, rate of climb, and fuel economy of the aircraft, adversely [6–9]. The relative positions of the centroid and the aerodynamic center are the key factors affecting the stability and maneuverability of an aircraft. Therefore, the first problem to be solved in morphing the aircraft design is balancing the influence of the change in the centroid and the aerodynamic center. At present, measures such as the deflection of the control surface or spread blowing on the wing surface are used to suppress the excessive movement of the aerodynamic center, but this limits the mission

adaptability of the morphing aircraft [10,11]. The large range changes in the centroid and moment of inertia are compensated by the use of lighter materials that meet the deformation requirements, or by the movement of internal fuel and slider [12–15].

With the progress of bionics research, the coordinate deformation mode of the sweep/dihedral angle of the arm wing and hand wing parts of large-sized birds provides a new approach for solving the above problems. By observing the wing shape change in raptor-sized birds, such as steppe eagle, swift, gull, albatross, etc., biologists have found that the arm wing (inboard section containing the secondary feathers) moves forward with the wrist joint, while the hand wing (outer section of the wing, consisting of the primary feathers) sweeps to the rear. The dihedral angles of the hand and arm wings also show an opposite change pattern [16–23]. The multi-section morphing wing can realize the centroid self-trim compensation during the morphing process by the antisymmetric coordinated deformation of the inner and outer wing sections. At present, the research on multi-section morphing wings mainly focuses on aerodynamic characteristics analysis, intelligent deformation structure design, and control system design. Marks et al. [24] have designed a four-bar mechanism to mimic the skeletal structure of a bird's wing. Parameters such as sweep angle, wing area, and wingspan are changed by folding deformation similar to that of the bird's feather stacking. A variable camber airfoil is used to replace the traditional split control surface to realize roll maneuver and landing flight control. Muhammad et al. [25] have designed a bionic folding wing. The wing structure is made of epoxy webs and polypropylene film. The deformation of the wings can be completed in the plane. Stowers et al. [26] have designed a folding wing that can be deployed in the plane through centrifugal acceleration, based on the study of the wing structure of birds and bats. Luca et al. [27] have designed a bionic folding wing that can be folded and deformed in the plane according to the feather structure and deformation law of bird wings. Through theoretical analysis and wind tunnel tests, the aerodynamic characteristics of a bionic wing composed of artificial feathers in different configurations have been studied, and the possibility of using the asymmetric sweep angle change in the outer wing section for roll maneuver control has been discussed. Grant et al. [28] have designed a multi-section variable-sweep unmanned aerial vehicle (UAV) that mimics a seagull wing. The inner and outer wing sections can independently change the sweep angle. The aerodynamic analysis of the aircraft shows that the symmetrical sweep angle change can significantly reduce the turning radius, and the asymmetric sweep angle change can enhance the ability to resist crosswind. The dynamics equations have been established, and the coupling relationship between longitudinal and lateral/directional during the asymmetric change in sweep angle has been studied. The influence of transient variables on the mission performance has been analyzed [29–31]. Hartloper and Wolf et al. [32,33] have studied the aerodynamic performance of a UAV imitating a seagull wing. Verstraete et al. [34] have used the unsteady vortex lattice method to establish a numerical calculation model for nonlinear and unsteady aerodynamic forces in the morphing process of a seagull wing. Obradovic et al. [35] have proposed a numerical method for calculating the dynamic loads and energy requirements of morphing wings during the morphing process, based on the vortex lattice method. At different deformation rates, angles of attack, and Mach numbers, Han, H. et al. [36] have used a numerical simulation method to analyze the unsteady aerodynamic characteristics of the UAV in the process of symmetrical sweep angle change in the outer wing section.

In summary, the current research on multi-section morphing wings mainly focuses on bird-sized wings with lightweight film materials and small wingspans, so the movement of the centroid and aerodynamic center, caused by configuration changes, is not prominent. There is a lack of research on the practical application of the multi-section variable-sweep wing in the high-aspect-ratio heavy-duty UAV. In addition, the relevant studies are all focused on a UAV with a specific wingspan of the inner and outer sections. There are also few studies data on the use of reverse coordinated deformation of the inner and outer sections to achieve centroid stability during the morphing process. The influence of

configuration changes on the aerodynamic characteristics and stability of heavy-duty UAVs has not been fully clarified, and there is a lack of systematic research on the coordinate trim compensation mechanism of multi-section morphing wings with a high aspect ratio. Under this morphing mode, the wingspan ratio of the inner and outer wing sections of the UAV will affect the design variables that determine the aircraft performance, such as the wingspan, aspect ratio, wing area, and sweep angle.

In response to the above problems, a design scheme of a bionic multi-section variable-sweep wing UAV that utilizes the reverse coordinated sweep angle change in the inner and outer wing sections is proposed, which improves the aerodynamic performance and realizes the centroid self-trim compensation. Based on the layout characteristics of this type of aircraft, a reasonable design of the wingspan ratio of the inner and outer wing sections is explored, to reduce the impact of aerodynamic center movement and the change in the moment of inertia, and to balance the contradiction among aerodynamic performance improvement, stability, and maneuverability of the morphing aircraft. Based on the morphing principle of the centroid self-trim compensation, this study analyzes the influence of multiple factors on the aerodynamic characteristics and stability of multi-section variable-sweep wing UAVs. Based on the analysis results, this study proposes a reasonable wingspan ratio of the inner and outer sections and provides a reference for the practical application of the multi-section variable-sweep wing in a high-aspect-ratio heavy-duty UAV. This paper is organized as follows. Section 2 presents the main theories and stability augmentation systems applied in this study. In Section 3, a bionic multi-section design scheme for a variable-sweep wing UAV is proposed. Based on the distribution characteristics of the wing structure, pre-selected models of UAVs with different wingspan ratios of the inner and outer sections are constructed. In Section 4, the variation in aerodynamic characteristics and longitudinal trim characteristics during the coordinate morphing process are analyzed by considering the pre-selected models as the research object with the centroid self-trim compensation. In Section 5, the influence of configuration change, and the wingspan ratio of the inner and outer sections on the stability of the targeted UAV under different working conditions is investigated.

## 2. Applied Theory

In this section, the main theories and stability augmentation systems applied in this study are presented.

### 2.1. Kane Dynamic Modeling Methods

An accurate and reasonable model is the basis for analyzing the dynamic response of multi-joint morphing UAVs. The assumptions of the single rigid-body model are no longer applicable to large-scale morphing UAVs. Therefore, a new dynamic model must be developed. Kane's methods choose a series of independent generalized velocities to describe the motion of the system. It is no longer necessary to calculate the energy function and related time derivatives to obtain the differential equations of the system without an ideal constraint reaction force [37]. Therefore, it is more suitable for targeted UAV. The specific modeling methods are as follows:

$f$ independent variables are selected as the system's generalized velocities, which are expressed as $u_k$ ($k = 1, 2, \ldots, f$). The translational velocity $v_{ci}$ and angular velocity $\omega_i$ of the $i$-th rigid body, relative to the centroid of the inertial reference system, can be separately expressed as a linear combination of the generalized velocities, as follows:

$$v_{ci} = \sum_{k=1}^{f} v_{ci}^{(k)} u_k + v_{ci}^{(0)}, \; \omega_i = \sum_{k=1}^{f} \omega_i^{(k)} u_k + \omega_i^{(0)} \tag{1}$$

where $v_{ci}(k)$, $v_{ci}(0)$, $\omega_i(k)$, $\omega_i(0)$ are functions of generalized velocities; $v_{ci}(k)$ is the $k$-th partial velocity of the centroid of the $i$-th rigid body; and $\omega_i(k)$ is the $k$-th partial angular

velocity of the centroid of the *i*-th rigid body. For a multi-body system with $N$ rigid bodies and $f$ degrees of freedom, Kane's equation can be expressed as follows:

$$F_k + F_k^* = 0, \quad k = 1, 2, \cdots, f \tag{2}$$

where $F_k$ and $F_k^{*}$ are the generalized active force and generalized inertial force corresponding to $u_k$, respectively.

$$F_k = \sum_{i=1}^{N} \left[ F_k \cdot v_{ci}^{(k)} + M_i \cdot \omega_i^{(k)} \right], \; F_k^* = \sum_{i=1}^{N} \left[ F_i^* \cdot v_{ci}^{(k)} + M_i^* \cdot \omega_i^{(k)} \right] \tag{3}$$

where $F_i$ and $F_i^{*}$ are the principal vectors of the active force and inertial force acting on the *i*-th rigid body, respectively; and $M_i$ and $M_i^{*}$ are the principal moments of the active force and inertial force, relative to the centroid of the *i*-th rigid body, respectively. They can be expressed as follows:

$$F_i^* = -m_i a_{ci}, \; M_i^* = -J_i \cdot \dot{\omega}_i - \omega_i \times (J_i \cdot \omega_i) \tag{4}$$

where $J_i$ is the inertial tensor and $a_{ci}$ is the acceleration of the centroid of the *i*-th rigid body. $\omega_i$ is the angular velocity of the *i*-th rigid body. If three perpendicular unit vectors, $c_1$, $c_2$, and $c_3$, form a right-handed system and are parallel to the central principal axis of the reference frame (the right-handed system is not required to be fixed), they can be defined as follows:

$$\left. \begin{array}{l} \dot{\omega}_j = \dot{\omega} \cdot c_j \\ \omega_j = \omega \cdot c_j \\ J_j = c_j \cdot J \cdot c_j \end{array} \right\}, \; (j = 1, 2, 3) \tag{5}$$

Then, the second equation of Equation (4) can be rewritten as follows:

$$\begin{aligned} T_i^* = \quad &- \left[ \dot{\omega}_{i1} \cdot J_{i1} - \omega_{i2} \omega_{i3} (J_{i2} - J_{i3}) \right] c_{i1} \\ &- \left[ \dot{\omega}_{i2} \cdot J_{i2} - \omega_{i3} \omega_{i1} (J_{i3} - J_{i1}) \right] c_{i2} \\ &- \left[ \dot{\omega}_{i3} \cdot J_{i3} - \omega_{i1} \omega_{i2} (J_{i1} - J_{i2}) \right] c_{i3} \end{aligned} \tag{6}$$

### 2.2. Design of Stability Augmentation Systems

The longitudinal stability augmentation system (SAS) can improve the characteristics of the oscillation frequency and damping of the short-period modal by feeding back the angle of attack and pitch rate to the elevator, thereby improving the flight quality of the aircraft over a wider range [38]. Figure 1 is the schematic diagram of the longitudinal stability augmentation system, in which the change in the angle of attack is felt by the sensor. After amplifying the signal by the system, the rudder control is deflected to reduce the change in the angle of attack through the steering gear and booster. The pitch rate sensor is usually a mechanical gyro device that measures the angular rate around the pitch axis.

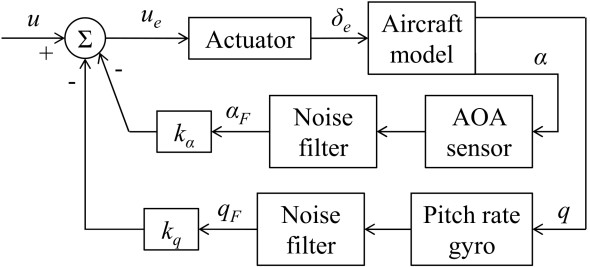

**Figure 1.** Longitudinal stability augmentation system. The characteristics of the oscillation frequency and damping of the short period modal are modified by feeding back the angle of attack and pitch rate to the elevator.

Figure 2 shows the lateral/directional stability augmentation system adopted in this study, in which the body-axis roll rate is fed back to the ailerons and sweep angle to modify the roll subsidence modal, and the yaw rate is fed back to the rudder on the V-tail to modify the Dutch roll mode. Generally, the lateral (roll) motion is coupled with yaw and sideslip (directional) motions. Therefore, the augmentation systems will be analyzed with the aid of multivariable state equations (three inputs, ailerons, rudder, and asymmetric sweep angle, and two or more outputs). The purpose of the stability augmentation yaw rate feedback is to use the rudder to generate a yaw moment that opposes any yaw rate that builds up from the Dutch roll mode. The resulting feedback configuration is a type 0 yaw rate command system with zero command input. This raises the following difficulty: in a coordinated steady-state turn, the yaw rate has a constant non-zero value. It may be thought that the pilot could easily coordinate the turn by applying the correct yaw rate command through the rudder pedals. A simple solution to the problem of implementing the yaw damper is to use a washout circuit on the output of the yaw rate sensor [38]. The high-pass filtering action of the washout circuit removes the steady-state component of the yaw rate during turns. The output of the washout approximates the differentiated yaw rate. In Figure 2, $G_w$ is the washout circuit, $G_a$ represents the transfer function of the control surface at the trailing edge of the outer wing part, $G_\Lambda$ represents the transfer function of the asymmetric sweep angle change in the outer wing part, and $G_r$ is the rudder actuator. The transfer functions $G_F$ represent noise filtering and any effective lag at the output of the roll rate and yaw rate gyros.

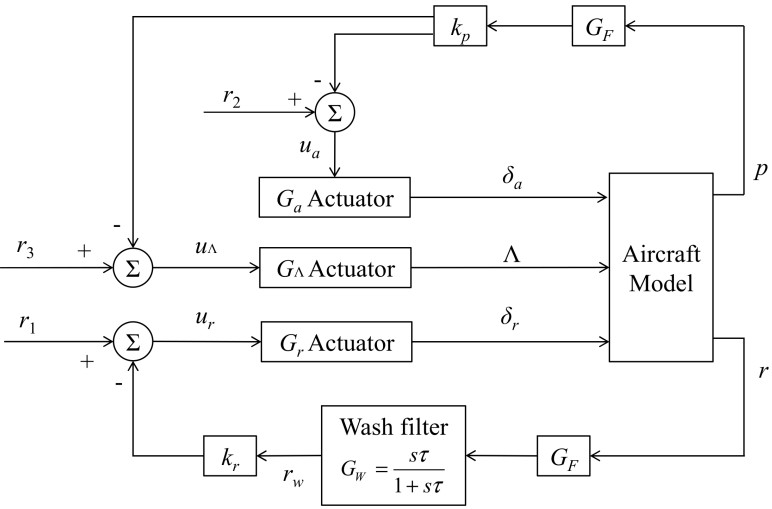

**Figure 2.** Lateral/directional augmentation system. The body-axis roll rate is fed back to the ailerons and sweep angle to modify the roll subsidence modal, and yaw rate is fed back to the rudder on the V-tail to modify the Dutch roll mode.

## 3. Overall Design of Bionic Multi-Section Variable-Sweep Wing UAV

The concept of a multi-section morphing wing based on bionics research provides an effective way to solve the problem of the centroid and stability control of high-aspect-ratio wings. In this section, a multi-section variable-sweep wing UAV design scheme is proposed, which uses the reverse coordination change in the sweep angle of the inner and outer wings to realize the centroid self-trim compensation in the process of wing configuration change.

### 3.1. Determination of Design Parameters of Targeted Morphing UAV

The high-aspect-ratio wing UAVs in active service are mostly in the range of medium and low subsonic speeds, with poor maneuverability, but good endurance performance [39,40]. The MQ-9 Reaper has a long range, long endurance, and large releasable payload-carrying capacity. It is the integrated UAV system with the largest amount of equipment and is the most widely used in the world today [41–43]. Taking MQ-9 as the design reference,

the UAV designed in this study adopts a conventional layout. The fuselage structure is slender and approximately cylindrical. The head is hemispherical, with built-in guidance, control equipment, and a mission load. The antenna is placed under the nose. The high-aspect-ratio multi-section variable-sweep wing is located in the middle of the fuselage. The wing can be divided into the following three sections: the weapon, and the inner and outer wings. The weapon section has underwing pylons. The centroid self-trim compensation for the configuration change in the UAV is realized by the reverse coordination deformation of the inner and outer wing sections. The trapezoidal V-tails are located at the rear of the fuselage with a tail fin. The lower fuselage has a retractable front three-point landing gear with external underwing pylons, and the main landing gear on either side of the centroid, as shown in Figure 3.

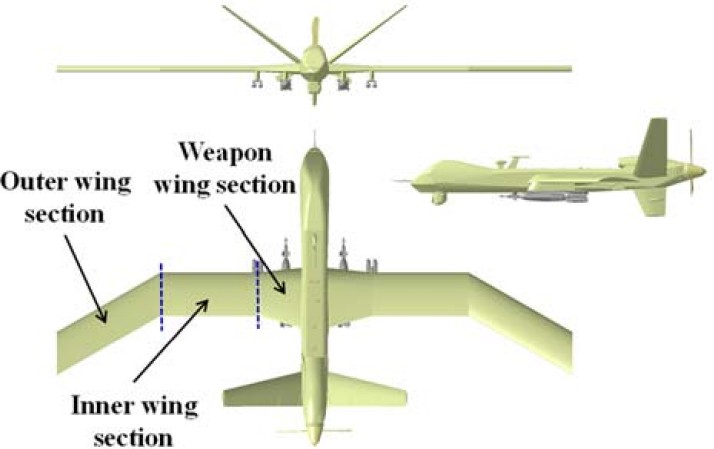

**Figure 3.** Schematic diagram of the bionic multi-section variable-sweep wing UAV with a conventional layout. The wing can be divided into the following three sections: weapon wing, inner wing, and outer wing.

Based on the above analysis and referring to the flight conditions of MQ-9, the flight performance parameters of the multi-section variable-sweep wing UAV designed in this study are determined as follows:

1. Cruising Mach number: Ma = 0.3, maximum Mach number: Ma = 0.6;
2. Cruising altitude: H = 9000 m;
3. In cruise flight, the maximum lift/drag ratio of the wing: $L/D$ > 20.

A shearing variable-sweep wing is chosen for the targeted morphing UAV. The wing adaptively changes the sweep angle if the flight state needs to be changed. The wing profile at each spanwise position is translated linearly in the flow direction, and the closer to the wingtip, the greater the translation. The wing tip is along the flow direction, and the airfoil on the windward side remains unchanged during the morphing process, which has little influence on the structure of the turbulence field [44,45].

Owing to the large mass of the weapon section, the change in its sweep angle has a significant influence on the stability of the UAV. Therefore, the sweep angle of the weapon section remains fixed during the wing deformation process. Based on the above considerations, the design parameters related to the multi-section wing, tail, and fuselage of the UAV in this research are determined, as shown in Table 1.

**Table 1.** Overall design parameters of wing, tail and fuselage.

| Parameters | Value |
|---|---|
| wing area | 32.15 m$^2$ |
| wingspan of weapon section | 1.91 m |
| sweep angle of weapon section | 6.1° |
| wingspan of deformable section | 8.09 m |
| wing-root length | 1.79 m |
| wingtip length | 1.59 m |
| taper of the morphing section | 1 |
| sweep angle of the wing | 6.1° |
| wingspan | 20 m |
| wing-root chord length | 1.79 m |
| fuselage length | 11 m |
| cross-sectional area of fuselage | 0.81 m$^2$ |
| equivalent diameter of fuselage | 1.01 m |
| sweep angel of V-tail | 13.7° |
| half-span length of V-tail | 4.13 m |
| root chord length of V-tail | 1.39 m |
| tip chord length of V-tail | 0.60 m |
| dihedral of V-tail | 36.66° |
| sweep angle of tail fin | 25.3° |
| half-span length of tail fin | 1.20 m |
| root chord length of tail fin | 1.31 m |
| tip chord length of tail fin | 0.73 m |

According to the weight estimation method in the overall design stage [46], the mass and centroid positions of the major components of the targeted morphing UAV studied in this research are listed in Table 2.

**Table 2.** Mass and centroid position of major components of targeted morphing UAV.

| Components | Mass/kg | Distance from Nose/m |
|---|---|---|
| fuselage | 530.47 | 4.5 |
| landing gear | 163.01 | 4.5 |
| spray paint | 23.08 | 4.5 |
| wing | 659.62 | 5.18 |
| tail | V-tail: 160.38 | 9.1 |
| | Tail fin: 15.61 | 8.97 |
| power unit | 506 | 8 |
| built-in load | 383 | 1.5 |
| fuel | 1180 | 4.2 |
| loaded ammunition | 1363 | 4.91 |
| total mass | 4981.16 | |

The multi-section variable-sweep wing UAV designed in this research aims to realize the centroid self-trim compensation during the morphing process by using the reverse coordination sweep angle change in the inner and outer sections. To more accurately describe the influence of sweep angle change on the wing structure distribution, and evaluate the influence of the wingspan ratio of the inner and outer sections on flight performance, the multi-section variable-sweep wing structure of the targeted morphing UAV needs to be designed before the aerodynamic calculation and stability analysis. The wing structure consists of four stringers that are distributed along the span direction, and are connected to the reinforcement frame of the fuselage at the wing root. The distribution of the wing ribs is related to the position of the control surface and landing gear. Based on this analysis, the number of stringers and ribs of the multi-section variable-sweep wing is determined to be 4 and 20, respectively, as shown in Figure 4. The positions and dimensions of the ribs from the wing root to the wing tip are listed in Table 3. The ribs No. 2 and

No. 4 are located in the positions of the underwing pylons. The structural weight gain coefficient of the deformation wing structure is determined to be 1.17. The weight gain is evenly distributed among the structural elements of the wing. The thickness of the wing skin is 2 mm.

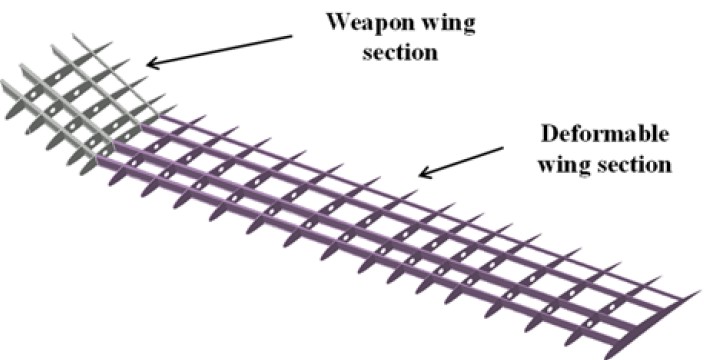

**Figure 4.** Shape and structure of multi-section variable-sweep wing. The gray part is the weapon section, and the purple part is the deformable section.

**Table 3.** Positions and dimensions of wing ribs.

| No | Spanwise Position/m | Chord Length/m | No | Spanwise Position/m | Chord Length/m |
|----|---------------------|----------------|----|---------------------|----------------|
| 1 | 0.434 | 1.741 | 11 | 5.053 | 1.588 |
| 2 | 0.834 | 1.700 | 12 | 5.626 | 1.588 |
| 3 | 1.243 | 1.658 | 13 | 6.199 | 1.588 |
| 4 | 1.652 | 1.615 | 14 | 6.645 | 1.588 |
| 5 | 1.911 | 1.588 | 15 | 7.091 | 1.588 |
| 6 | 2.360 | 1.588 | 16 | 7.537 | 1.588 |
| 7 | 2.859 | 1.588 | 17 | 8.153 | 1.588 |
| 8 | 3.358 | 1.588 | 18 | 8.769 | 1.588 |
| 9 | 3.908 | 1.588 | 19 | 9.384 | 1.588 |
| 10 | 4.480 | 1.588 | 20 | 10.000 | 1.588 |

*3.2. Coordinated Wing Deformation Scheme Based on the Centroid Self-Trim Compensation*

According to the distribution of wing ribs along the span direction, and considering the load-bearing capacity and the space required for wing deformation, pre-selected models of multi-section variable-sweep wings are established. After preliminary screening, the rib numbers at the boundary of the inner and outer wing sections of the pre-selected model are determined to be 12, 13, 14, 15, and 16, respectively. The five pre-selected models are named Model1, Model2, Model3, Model4, and Model5, respectively. The configurations with the sweep angle of the inner section $\Lambda_{in} = 0°$ and the outer section $\Lambda_{out} = 30°$ of the pre-selected models are shown in Figure 5. The mass and dimensions of the inner and outer sections of each pre-selected model are listed in Table 4.

The proportion of the wingspan of the inner section increases gradually

Model1 Model2 Model3 Model4 Model5

**Figure 5.** Schematic diagram of the dimensions of the inner and outer sections corresponding to the pre-selected models. The sweep angle of the inner section $\Lambda_{in} = 0°$ and outer section $\Lambda_{out} = 30°$.

**Table 4.** Mass and dimensions of the inner and outer sections of the pre-selected models.

| Model | Mass of Inner Section/kg | Mass of Outer Section/kg | Wingspan of Inner Section/m | Wingspan of Outer Section/m |
|---|---|---|---|---|
| Model1 | 132.21 | 89.99 | 3.72 | 4.37 |
| Model2 | 154.66 | 67.55 | 4.29 | 3.80 |
| Model3 | 158.43 | 63.77 | 4.73 | 3.39 |
| Model4 | 168.88 | 53.33 | 5.18 | 2.91 |
| Model5 | 178.65 | 43.55 | 5.63 | 2.46 |

Figure 6 shows the schematic diagram of the deformation process of the wing deformation mode for the multi-section variable-sweep wing UAV based on the centroid self-trim compensation, in which the sweep angle of the inner section $\Lambda_{in}$ gradually increases from $0°$, while the sweep angle of the outer section $\Lambda_{out}$ decreases to $0°$.

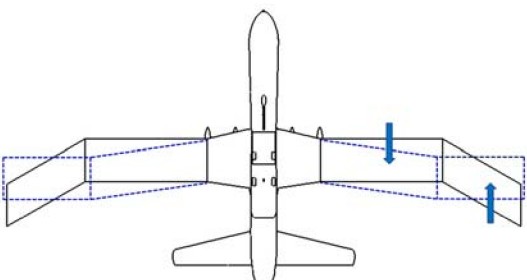

**Figure 6.** Schematic diagram of the wing deformation mode for multi-section variable-sweep wing UAV based on the centroid self-trim compensation, in which the sweep angle of the inner section gradually increases from $0°$ while the sweep angle of the outer section decreases to $0°$.

Equation (7) lists the fitting functions of the sweep angle change in each wing section of the pre-selected models based on the centroid self-trim compensation.

$$
\begin{aligned}
y_{\text{Model1}} &= 0.003635x^2 - 0.8632x + 22.61 \\
y_{\text{Model2}} &= 0.00133x^2 - 0.405x + 10.93 \\
y_{\text{Model3}} &= 0.0009565x^2 - 0.3035x + 8.214 \\
y_{\text{Model4}} &= 0.0005743x^2 - 0.1917x + 5.222 \\
y_{\text{Model5}} &= 0.0003501x^2 - 0.1184x + 3.229
\end{aligned}
\tag{7}
$$

where $x$ is the sweep angle of the outer section, and $y$ is the sweep angle of the inner section. The unit is in degrees ($°$).

*3.3. Influence Analysis of Static Stability*

Static stability is mainly related to the aerodynamic layout of an aircraft. After the overall parameters are determined, some factors related to dynamic stability are still unclear. Therefore, the static stability is initially used to measure the stability of the aircraft.

If the sweep angles of the inner and outer sections change synchronously, the multi-section variable-sweep wing is transformed into a traditional single-section variable-sweep wing. Table 5 lists the longitudinal static stability margins of the single-section wing UAV under different sweep angles. This research also considers the situation in which the wing installation position moves 0.1 m backwards, which can simulate the initial state of the single-section wing with different sweep angles, to a certain extent. As shown in Table 5, the change in the sweep angle of the traditional single-section wing has a significant influence on the longitudinal static stability margin of the high-aspect-ratio wing aircraft. In the absence of other measures, it is not suitable for this type of aircraft to cause a large change in the sweep angle. However, this problem can be solved by the centroid self-trim compensation of a multi-section variable-sweep wing.

**Table 5.** The value of longitudinal static stability margin at different sweep angles of a single-section wing UAV.

| Sweep Angle/° | Weapon Mounted | No Weapon Mounted | Releasable-Payload + Wing Backward 0.1 m | Without Releasable-Payload + Wing Backward 0.1 m |
|---|---|---|---|---|
| 0 | −5.43% | −7.96% | −0.13% | −2.97% |
| 5 | −14.50% | −16.43% | −9.20% | −11.44% |
| 10 | −23.51% | −24.86% | −18.22% | −19.87% |
| 15 | −32.41% | −33.18% | −27.11% | −28.19% |
| 20 | −41.12% | −41.35% | −35.83% | −36.36% |
| 25 | −49.61% | −49.31% | −44.31% | −44.33% |
| 30 | −57.80% | −57.03% | −52.51% | −52.04% |

According to the coordinated sweep angle change in the inner and outer sections of the pre-selected models, under the morphing mode of the centroid self-trim compensation, the variation in the longitudinal static stability margin of the UAV with the sweep angle change in the outer section corresponding to the configuration can be obtained, as shown in Figure 7. As can be observed from Figure 7, when the wingspan of the inner section is relatively small, the variation in $\Lambda_{in}$, caused by the change in $\Lambda_{out}$, is relatively large. In the coordinate sweep angle change process of the wing sections, the aerodynamic center moves forward gradually, which causes the static stability margin of the aircraft to decrease with the increase in $\Lambda_{out}$. As the proportion of the wingspan of the inner section of the pre-selected models increases, the variation in $\Lambda_{in}$, caused by $\Lambda_{out}$, decreases, and the forward movement of the aerodynamic center gradually decreases, so the decreasing speed of the static stability margin gradually decreases. With a further increase in the wingspan proportion of the inner section, the backward movement of the aerodynamic center, caused by the increase in $\Lambda_{out}$, is greater than the forward movement of the aerodynamic center, caused by the decrease in $\Lambda_{in}$. The longitudinal static stability margin decreases with the increase in $\Lambda_{out}$, and the change rate of the static stability margin shows a gradually increasing trend with the increase in $\Lambda_{out}$. In addition, it can be observed that when the UAV does not have releasable payload, the mass of the wing is reduced and the centroid moves forward. Therefore, the longitudinal static stability margin under this condition is greater than that when the UAV has a releasable payload.

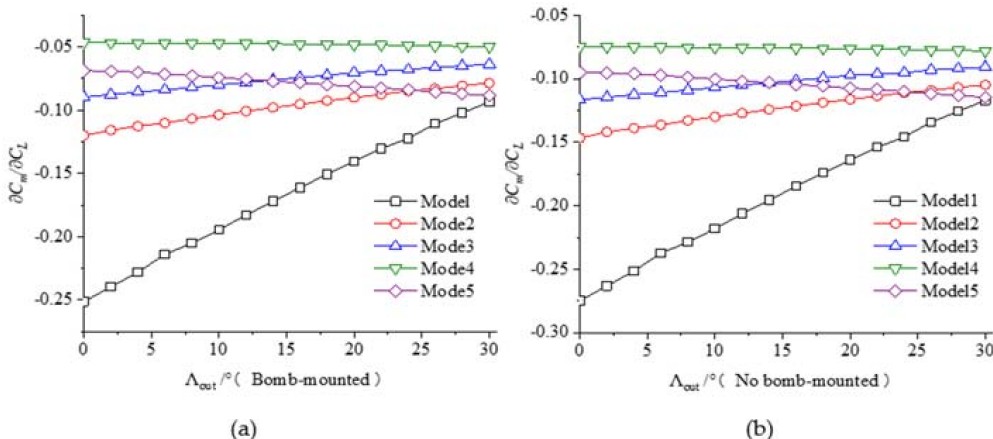

**Figure 7.** The variation in the longitudinal static stability margin of the multi-section variable-sweep UAV with the sweep angle of the outer section based on the centroid self-trim compensation. (**a**) Releasable payload, (**b**) without releasable payload.

For the multi-section variable-sweep wing UAV, it is hoped that the influence of configuration changes on the static stability margin will be minimized as much as possible. The following conclusions can be drawn from the calculation and analysis results:

1.  With reducing the mass of components near the centroid, such as the increase in fuel consumption, the static stability margin of the multi-section variable-sweep wing UAV will gradually increase;

2.  From the perspective of the longitudinal static stability margin changes during the coordinated deformation of the inner and outer sections of the pre-selected models, the static stability characteristic of Model2~Model5 is significantly better than Model1. The variation range of the static stability margin of Model3~Model5 meets the design requirements. Among them, the static stability margin of Model4 has the smallest variation range, with a releasable payload, as follows: CM= −4.98~−4.66%, and without a releasable payload, as follows: CM = −7.80~−7.48%. At this time, the wingspan ratio of the inner and outer sections is approximately 1.78.

## 4. Aerodynamic Analysis of UAV Based on the Centroid Self-Trim Deformation

The multi-section variable-sweep wing UAV can realize the centroid self-trim compensation through the coordinated sweep angle change in the inner and outer wing sections. Compared with the traditional single-section variable-sweep wing, this special deformation method presents new laws for changes in wing aerodynamic characteristics. In addition, for aircrafts with different wingspan ratios of the inner and outer sections, the aerodynamic characteristics are different, owing to the different ranges of parameters of the wing sections of pre-selected models, such as sweep angle, wingspan, and wing area, etc. Therefore, the aerodynamic analysis of the pre-selected models, based on the coordinated deformation mode, is performed using Fluent. The flow control equation is Reynolds average NS equation, and the turbulence model adopts S-A (Spalart–Allmaras) model. Subsequently, a multivariable influence analysis is carried out on the longitudinal pitch trim characteristics.

### 4.1. Multivariable Influence Analysis on Aerodynamic Characteristics of UAV

According to the flight conditions of the multi-section variable-sweep wing UAV, the aerodynamic characteristics of the four pre-selected models are analyzed within the following range of angle of attack: AOA = 0~10°, and the following Mach number: Ma = 0.3~0.6.

Figure 8 shows the variation in the lift coefficient $C_L$ and drag coefficient $C_D$ of Model2 at different Mach numbers, with the sweep angle change in the outer section $\Lambda_{out}$ under the coordinate morphing mode when AOA = 0°. The variations in the lift and drag coefficients at other angles of attack are similar to this condition. The following conclusions are drawn:

1. As the $\Lambda$out corresponding to the configuration increases, $C_L$ and $C_D$ both show the law of change that first increases slightly and then gradually decreases. When the sweep angle of the outer section varies in the range of 0~10°, the aerodynamic changes generated by the inner and outer wing sections in the coordinated deformation mode are similar. The total aerodynamic force of the wing changes slightly.
2. For a given wing configuration, $C_D$ first decreases and then increases with an increase in the flight speed. During the coordinate deformation process, the minimum drag coefficient $C_{D\text{min}}$ of the UAV corresponds to the configuration $\Lambda_{\text{in}} = 0°$ and $\Lambda_{\text{out}} = 30°$.

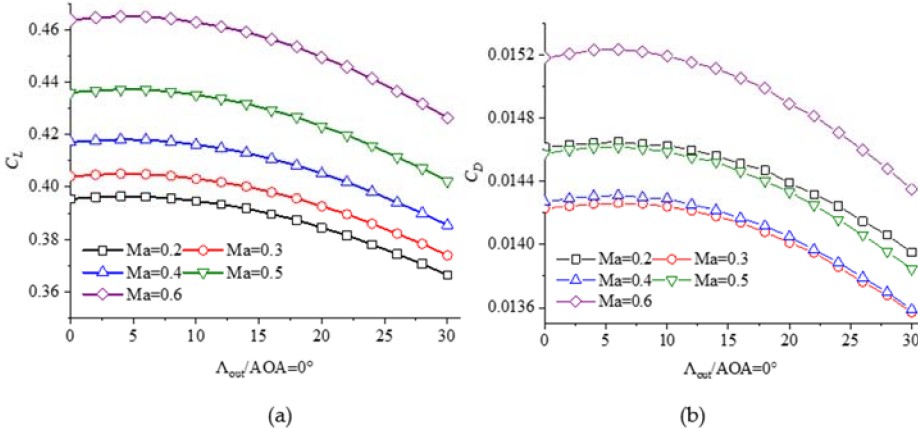

**Figure 8.** Aerodynamic coefficient changes with the configuration of Model2 at different Mach numbers with the sweep angle change in outer section under the coordinate morphing mode when AOA = 0°. (**a**) Lift coefficient, (**b**) drag coefficient.

Figure 9 shows the variation in the lift line slope $C_{L\alpha}$ of the pre-selected models, with the sweep angle of the outer section $\Lambda_{\text{out}}$ corresponding to the configuration when the angle of attack is AOA = 0° and the Mach number is Ma = 0.3. Similar laws can be obtained for other angles of attack and Mach numbers. The following conclusions are drawn:

1. With an increase in the wingspan proportion of the inner section, the slope of the lift line gradually increases. The slope of the lift line of Model2 is significantly smaller than that of the other three models. When the sweep angle of the outer section, corresponding to the configuration, changes within the range of 0~10°, the numerical difference in the slope of the lift line of Model3~Model5 is small;
2. When the wingspan proportion of the inner section is small, the slope of the lift line increases first and then decreases with an increase in the $\Lambda_{\text{out}}$ corresponding to the configuration. This shows that when $\Lambda_{\text{out}}$ is small, the lift decrease caused by the increase in $\Lambda_{\text{out}}$ is smaller than the lift increase caused by the decrease in $\Lambda_{\text{in}}$. Overall, the total lift shows a slightly increasing trend. When $\Lambda_{\text{out}}$ is large, the lift loss caused by the sweep angle change in the outer section is not sufficient to be compensated by the lift generated by the sweep angle change in the inner section, so that the slope of the lift line decreases;
3. For a model with a large wingspan proportion of the inner section, the change in $\Lambda_{\text{in,}}$ caused by the change in $\Lambda_{\text{out}}$, is relatively small. The lift decrease in the outer wing section is greater than the increase in the inner wing section. The slope of the lift line decreases monotonically, and the decrease in speed gradually increases.

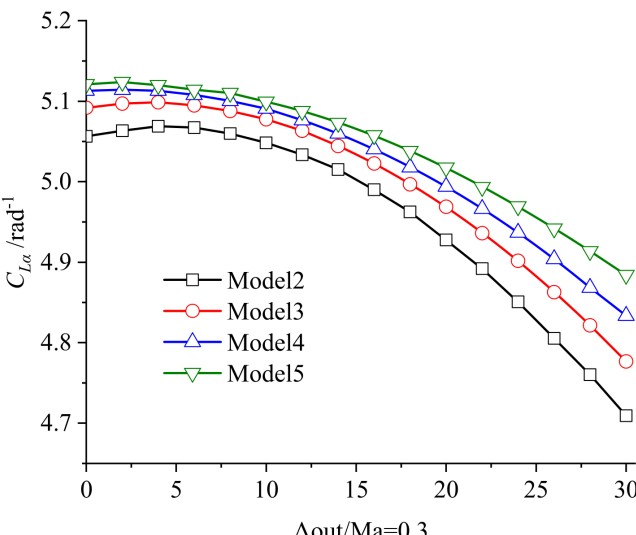

**Figure 9.** The variation in the lift line slope of the pre-selected models when AOA = 0° and Ma = 0.3.

Figure 10 shows the variation in the lift/drag ratio $L/D$ of the pre-selected models, with the $\Lambda_{out}$ corresponding to the configuration when Ma = 0.3. The angles of attack listed in the figures are 0°, 6°, and 10°, respectively. The change in the lift/drag ratio of the pre-selected models under other Mach numbers is similar to this situation. The following conclusions can be drawn:

1. With the increase in AOA, the value of $L/D$ of the pre-selected models first increases and then decreases. When the AOA is small, the value of $L/D$ of Model2 is significantly smaller than that of the other models. When the AOA is relatively large, Model2 has the largest value of $L/D$ when the sweep angle of the outer section, corresponding to the configuration, is small;

2. When the $\Lambda_{out}$ corresponding to the configuration increases, the difference in $L/D$ between different pre-selected models presents a changing law that first decreases and then increases. When the sweep angle of the outer section, corresponding to the configuration range from 0° to 10°, the difference in $L/D$ between Model4 and Model5 under different AOAs is small;

3. When the wingspan proportion of the inner section of the pre-selected model is relatively small, the value of $L/D$ first increases and then decreases with an increase in $\Lambda_{out}$, corresponding to the configuration. When the wingspan proportion of the inner section is relatively large, the value of $L/D$ gradually decreases with an increase in $\Lambda_{out}$, corresponding to the configuration.

Considering that the wingspan ratio of the inner and outer sections affects the efficiency of the control surface, and integrating the change in the lift and drag coefficients of the pre-selected models under different angles of attack, Model3 and Model4 have better aerodynamic performance.

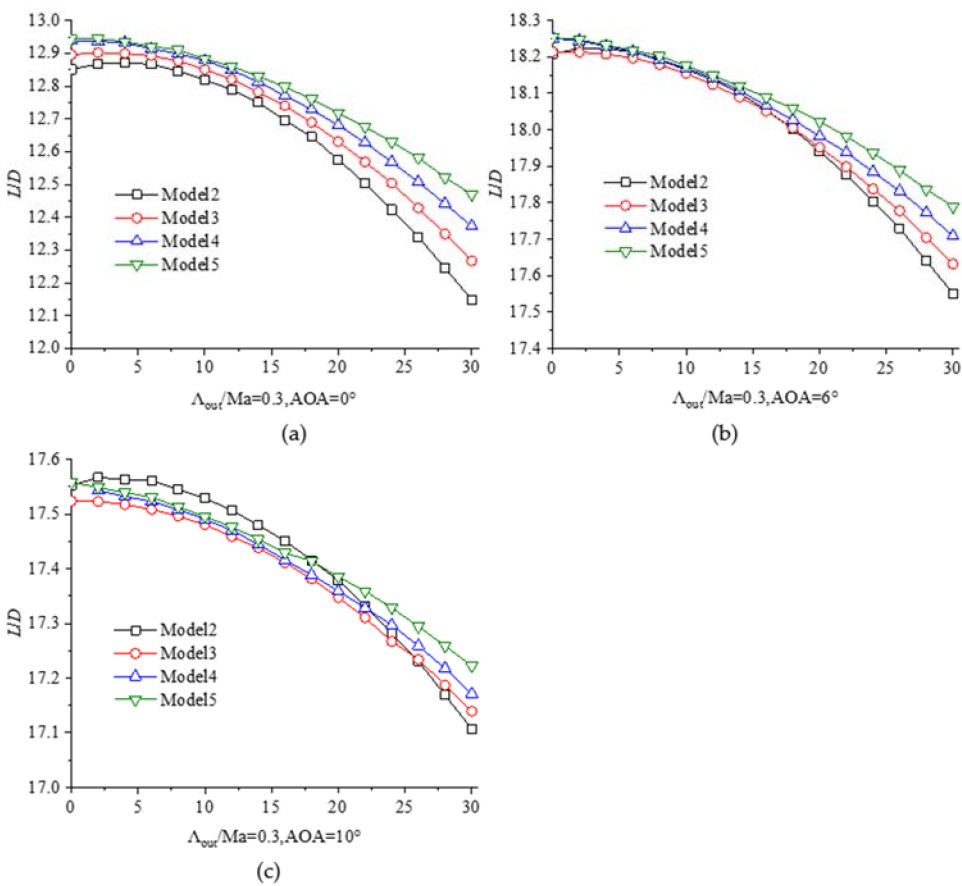

**Figure 10.** The change in the lift/drag ratio $L/D$ of the pre-selected model under different angles of attack when Ma = 0.3. The angles of attack equals the following: (**a**) AOA = $0°$, (**b**) AOA = $6°$, (**c**) AOA = $10°$, respectively.

### 4.2. Multivariable Influence Analysis on Pitching Trim Characteritics of UAV

The wing configuration is symmetrical when the UAV is in a straight and level flight. In this condition, the UAV is not subject to any external interference and only needs to be trimmed for the longitudinal pitch motion at this time. This section explores the influence of the configuration and distribution of the wingspan ratio of the inner and outer sections on the trim status of the UAV.

This research considers the influence of three main parameters, the angle of attack $\alpha$, the equivalent elevator deflection angle of the tail $\delta_e$, and the engine thrust $T$ on the trim results. The resultant force and moment of the UAV are equal to zero under the combined influence of the three parameters. According to the aerodynamic calculation results, the lift of the wing and tail have a linear relationship with the angle of attack, and the drag has a quadratic polynomial relationship with the angle of attack. The increments of $\Delta C_L$ and $\Delta C_D$ produced by the elevator have a quadratic polynomial relationship with the deflection angle. Therefore, for the longitudinal pitch motion, the following relationship needs to be satisfied at the trim point:

$$
\begin{aligned}
T\cos\alpha + \tfrac{1}{2}\rho V^2 S(C_{D0} + C_{D,\alpha2}\alpha^2 + C_{D,\alpha1}\alpha) &= 0 \\
T\sin\alpha + \tfrac{1}{2}\rho V^2 S(C_{L0} + C_{L,\alpha}\alpha + C_{L,\delta_e2}\delta_e^2 + C_{L,\delta_e1}\delta_e) - G &= 0 \\
\tfrac{1}{2}\rho V^2 S(C_{m0} + C_{m\alpha1}\alpha^2 + C_{m\alpha2}\alpha + C_{m,\delta_e2}\delta_e^2 + C_{m,\delta_e1}\delta_e) &= 0
\end{aligned}
\tag{8}
$$

where $\rho$ represents the air density, $V$ represents the flight speed, $S$ represents the reference area, and $G$ represents the weight of the aircraft. The direction of the engine thrust $T$ is along the longitudinal axis of the fuselage. Because of the smaller wing area of the V-tail, the influence of the increase in drag is ignored, and only the effect of the increase in lift and

the increase in pitch moment caused by its deflection is considered in the longitudinal trim modeling.

Table 6 shows the state value of the trim, corresponding to some configurations of the pre-selected models when Ma = 0.3. The following can be observed from the trim results:

1.  With the increase in $\Lambda_{\text{out}}$, corresponding to the configuration, the angle of attack $\alpha_{\text{trim}}$ first decreases and then increases, and the engine thrust $T_{\text{trim}}$ first increases and then decreases. The configurations corresponding to the minimum drag coefficient $C_{D\min}$ have the largest value of $\alpha_{\text{trim}}$ and the smallest value of $\delta_{e,\text{trim}}$;

2.  The state value of $\delta_{e,\text{trim}}$ increases approximately linearly with an increase in $\Lambda_{\text{out}}$, corresponding to the configuration. With an increase in the wingspan length of the inner section, the value of $\delta_{e,\text{trim}}$ gradually decreases, and the slope of the deflection curve of the rudder also gradually decreases;

3.  As the wingspan proportion of the inner section of the pre-selected model increases, the state value of $T_{\text{trim}}$ gradually decreases. Because $\Lambda_{\text{in}}$, corresponding to the same change in $\Lambda_{\text{out}}$, gradually decreases, the total lift of the UAV increases and the corresponding state value of $\alpha_{\text{trim}}$ gradually decreases. When $\Lambda_{\text{out}}$, corresponding to the configuration, changes from $0°$ to $10°$, the aerodynamic change in the UAV is relatively gentle. Among them, the state value range of $\alpha_{\text{trim}}$, corresponding to Model4 and Model5, is relatively small;

4.  With an increase in the wingspan proportion of the inner section, the engine thrust in the trim state gradually decreases.

In addition, it can be observed, from the longitudinal trim results, that an appropriate increase in the wingspan proportion of the inner section can reduce the trim resistance of the tail, reduce the engine thrust, and increase the range and endurance simultaneously.

**Table 6.** The trim state corresponding to some configurations of the pre-selected models when Ma = 0.3.

| $\Lambda_{\text{out}}/°$ | Model2 | | | Model3 | | |
|---|---|---|---|---|---|---|
| | $\alpha_{\text{trim}}/°$ | $\delta_{e,\text{trim}}/°$ | $T_{\text{trim}}/N$ | $\alpha_{\text{trim}}/°$ | $\delta_{e,\text{trim}}/°$ | $T_{\text{trim}}/N$ |
| 30 | 5.19 | −10.34 | 2889.88 | 5.01 | −9.68 | 2866.33 |
| 24 | 4.97 | −10.58 | 2876.28 | 4.82 | −9.9 | 2858.97 |
| 18 | 4.8 | −10.84 | 2868.34 | 4.69 | −10.16 | 2852.93 |
| 12 | 4.72 | −11.17 | 2866.37 | 4.61 | −10.46 | 2851.53 |
| 6 | 4.7 | −11.53 | 2867.35 | 4.59 | −10.8 | 2852.47 |
| 0 | 4.77 | −12.03 | 2877.85 | 4.64 | −11.22 | 2860.82 |

| $\Lambda_{\text{out}}/°$ | Model4 | | | Model5 | | |
|---|---|---|---|---|---|---|
| | $\alpha_{\text{trim}}/°$ | $\delta_{e,\text{trim}}/°$ | $T_{\text{trim}}/N$ | $\alpha_{\text{trim}}/°$ | $\delta_{e,\text{trim}}/°$ | $T_{\text{trim}}/N$ |
| 30 | 4.86 | −9.18 | 2850.68 | 4.74 | −8.84 | 2836.45 |
| 24 | 4.71 | −9.37 | 2840.49 | 4.62 | −9.02 | 2827.38 |
| 18 | 4.59 | −9.59 | 2836.93 | 4.53 | −9.2 | 2825.31 |
| 12 | 4.52 | −9.82 | 2832.02 | 4.47 | −9.41 | 2822.04 |
| 6 | 4.5 | −10.08 | 2834.75 | 4.45 | −9.63 | 2821.76 |
| 0 | 4.52 | −10.41 | 2835.16 | 4.46 | −9.88 | 2822.96 |

## 5. Stability Analysis and Augmentation Adjustment of UAV

When the targeted UAV designed in this research is morphing, the parameters affecting the stability of the aircraft, such as the centroid, moment of inertia, wingspan, aerodynamic chord length, and aerodynamic coefficient, change constantly. Especially for a multi-section variable-sweep UAV with high-aspect-ratio wings, the complex wing deformation causes the relative position of each wing section to change at all times. There is a serious coupling between the aircraft shape design parameters and aerodynamic performance; therefore, the changes in dynamic stability are also complicated. In addition, the dynamics equation based on Kane's method should consider the influence of structural

deformation on the centroid of the wing sections. In this section, the pre-selected models are considered as the research object, and the stability augmentation adjustment is carried out based on the flight quality requirement. Finally, the longitudinal and lateral/directional modal characteristics of the targeted multi-section variable-sweep wing UAV are calculated and analyzed.

### 5.1. Multi-Section Variable-Sweep Wing UAV System

The research object of this study can be considered as a four-joint, four-angle variable-sweep wing UAV system. Based on its overall layout, it can be regarded as a multi-body system that is composed of the following five main sections: left inner wing $W_1$, left outer wing $W_2$, right inner wing $W_3$, right outer wing $W_4$, and fuselage *FL* (including engine, tail, wing weapon section, etc.). Because the masses of other movable parts, such as the trailing edge of the tail and the control surface of the wing, are small, they are not treated as separate rigid bodies, and only the force or moment generated by the control surface deflection is considered. $W_1$ and $W_3$ are hinged to *FL* at $O_1$ and $O_3$, respectively, and $W_2$ and $W_4$ are hinged on $W_1$ and $W_3$ at $O_2$ and $O_4$, respectively, as shown in Figure 11. $O_f$ is the centroid of *FL*, and the sweep angles of each wing-body are defined as $\Lambda_1$, $\Lambda_2$, $\Lambda_3$, and $\Lambda_4$, respectively. In this research, the coordinates $O_f$, the centroid of the *FL*, are chosen as the origin, to establish the body coordinate system. The $O_x$ axis is in the symmetrical plane of the UAV, pointing forward, parallel to the axis of the fuselage; the $O_z$ axis is located in the plane of symmetry, perpendicular to the $O_x$ axis; and the $O_y$ axis is perpendicular to the symmetry plane and points to the right. In this research, wind speed is not considered in the modeling, that is, it is assumed to be carried out in a static wind field.

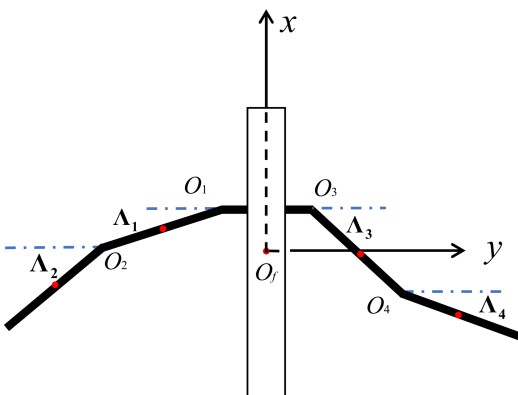

**Figure 11.** Schematic diagram of the multi-joint variable-sweep UAV system. The fuselage, tail, and weapon section are treated as one rigid body.

According to the dynamic modeling methods described in Section 2.1, the coordinates of $O_f$ in the Earth coordinate system and the projection of pitch, yaw, and roll angles, along with the sweep angle for each wing part in the body coordinate system, are selected as generalized coordinates, which are defined as $(x,y,z,\theta,\psi,\Phi,\Lambda_1,\Lambda_2,\Lambda_3,\Lambda_4)$. The projection of the translational velocity of $O_f$ and the angular velocity of rotation on the body coordinate system, and the change rate of the sweep angle of each body are selected as the system's generalized velocity, which is defined as $(u,v,w,p,q,r,\eta_1,\eta_2,\eta_3,\eta_4)$. To simplify the dynamic model, the following reasonable assumptions are made:

1. Ignoring the rotation and angular velocity of the Earth, and the Earth is considered to be stationary;
2. Ignoring the curvature of the Earth's surface, and the Earth coordinate system is considered as the inertial coordinate system;
3. Ignoring the changes in the fuel weight during flight, i.e., the aircraft's weight is always fixed.

### 5.2. Establishment and Linearization of Dynamic Equations

The dynamics equations of the multi-section variable-sweep wing UAV are established based on Kane's method. The influence of the centroid offset, caused by the wing structure deformation, is considered to modify the relative terms of the wing sections in the dynamics model. The dynamics model of the targeted morphing UAV system can then be simplified as follows:

$$J_E E + J_A \begin{bmatrix} L_w \\ D_w \\ L_t \\ D_t \\ Y_t \\ D_f \end{bmatrix} + J_T T + E_G \begin{bmatrix} G_f \\ G_w \end{bmatrix} + \begin{bmatrix} 0 \\ \vdots \\ L_p \bar{p} \\ \vdots \\ 0 \end{bmatrix} = K_V \begin{bmatrix} \dot{u} \\ \dot{v} \\ \dot{w} \end{bmatrix} + K_\Omega \begin{bmatrix} \dot{p} \\ \dot{q} \\ \dot{r} \end{bmatrix} + K_\eta \dot{\eta} + N \tag{9}$$

where $J_E$ represents the coefficient matrix related to engine thrust; $J_A$ represents the coefficient matrix related to the aerodynamics of each wing section, V-tail, and fuselage; $J_T$ represents the coefficient matrix related to the torque at each wing section; $E_G$ represents the coefficient matrix related to the mass in which the tail and fuselage are treated as a body; $L_p \bar{p}$ represents the effect of roll damping; $K_V$ represents the coefficient matrix related to velocity; $K_\Omega$ represents the coefficient matrix related to the angular velocity of the fuselage; $K_\eta$ represents the coefficient matrix related to the angular acceleration of sweep change; and $N$ contains the square term and the cross term of the generalized velocity of the UAV. Among them, the correction term of the centroid offset appears in the relevant positions of $J_A$, $K_V$, $K_\Omega$, and $K_\eta$, respectively, to achieve the modification of aerodynamic force, velocity, angular velocity, and wing deformation.

Considering the difference between the centroid of the wing with the actual structure, and that of the wing with uniform distribution, and the movement of the centroid of the wing in the $Ox_b y_b$ plane, caused by the deformation of the structure, the modified values of the centroid offset of the inner and outer sections are introduced into the model, which are defined as ($\triangle x_{in}, \triangle y_{in}, 0$) and ($\triangle x_{out}, \triangle y_{out}, 0$). The modified values are functions related to the actual structure distribution and deformation mode. The modified offset of the centroid can be expressed as a quadratic function of the sweep angle of the section when the wingspan ratio of the inner and outer sections is determined. Table 7 lists the correction functions of the centroid offset of the right-wing sections, considering the influence of the actual structure deformation of the wing, in which the independent variable is the sweep angle corresponding to the wing sections of each configuration. The correction functions of the centroid offset of the inner and outer wings of the left wing can be obtained using the corresponding coordinate transformation.

In this analysis, the thrust of the engine is along the axis of the fuselage. According to the previous analysis, it can be observed that the trimming angle of attack is small, so it can be considered that $\alpha^* = 0°$. Based on the small disturbance hypothesis [38], the linearized results of the lift, drag, and lateral forces can be expressed as follows:

$$\begin{aligned} \Delta D &= D_V \Delta V + D_\alpha \Delta \alpha + D_\beta \Delta \beta + D_{\delta_a} \Delta \delta_a + D_{\delta_e} \Delta \delta_e + D_\Lambda \Delta \Lambda \\ \Delta L &= L_V \Delta V + L_\alpha \Delta \alpha + L_{\dot{\alpha}} \Delta \dot{\alpha} + L_\beta \Delta \beta + L_q \Delta q + L_{\delta_a} \Delta \delta_a + L_{\delta_e} \Delta \delta_e + L_\Lambda \Delta \Lambda \\ \Delta Y &= Y_\beta \Delta \beta + Y_p \Delta p + Y_\gamma \Delta \gamma + Y_{\delta_r} \Delta \delta_r \end{aligned} \tag{10}$$

where $\triangle \delta_e$ and $\triangle \delta_r$ represent the equivalent deflection angles of the flat elevator and rudder produced by the V-tail, respectively; $\triangle \delta_a$ represents the deflection angle of the aileron at the trailing edge of the outer section; and $\triangle \Lambda$ represents the change in the sweep angle of the corresponding wing sections.

**Table 7.** The modification functions of the centroid offset of pre-selected model sections considering the actual structure.

| Pre-Selected Model | Correction Function of Centroid Offset |
|---|---|
| Model2 | $\triangle x_{\text{in}} = 7.52 \times 10^{-6}\Lambda^2 + 0.0039\Lambda + 0.1520$ <br> $\triangle y_{\text{in}} = 2.31 \times 10^{-5}\Lambda^2 - 1.86 \times 10^{-5}\Lambda + 0.2226$ <br> $\triangle x_{\text{out}} = 1.52 \times 10^{-5}\Lambda^2 + 0.0035\Lambda + 0.1040$ <br> $\triangle y_{\text{out}} = 1.39 \times 10^{-5}\Lambda^2 - 2.29 \times 10^{-5}\Lambda - 0.2086$ |
| Model3 | $\triangle x_{\text{in}} = 7.76 \times 10^{-6}\Lambda^2 + 0.0046\Lambda + 0.1503$ <br> $\triangle y_{\text{in}} = 2.70 \times 10^{-5}\Lambda^2 - 2.14 \times 10^{-5}\Lambda - 0.2627$ <br> $\triangle x_{\text{out}} = 1.41 \times 10^{-5}\Lambda^2 + 0.0027\Lambda + 0.0997$ <br> $\triangle y_{\text{out}} = 1.07 \times 10^{-5}\Lambda^2 - 2.29 \times 10^{-5}\Lambda - 0.1632$ |
| Model4 | $\triangle x_{\text{in}} = 8.48 \times 10^{-6}\Lambda^2 + 0.0054\Lambda + 0.1486$ <br> $\triangle y_{\text{in}} = 3.16 \times 10^{-5}\Lambda^2 - 3.21 \times 10^{-5}\Lambda - 0.3095$ <br> $\triangle x_{\text{out}} = 1.28 \times 10^{-5}\Lambda^2 + 0.0020\Lambda + 0.0948$ <br> $\triangle y_{\text{out}} = 7.91 \times 10^{-6}\Lambda^2 - 1.79 \times 10^{-5}\Lambda - 0.1196$ |
| Model5 | $\triangle x_{\text{in}} = 9.19 \times 10^{-6}\Lambda^2 + 0.0063\Lambda + 0.1467$ <br> $\triangle y_{\text{in}} = 3.61 \times 10^{-5}\Lambda^2 - 2.79 \times 10^{-5}\Lambda - 0.3632$ <br> $\triangle x_{\text{out}} = 1.20 \times 10^{-5}\Lambda^2 + 0.0013\Lambda + 0.0901$ <br> $\triangle y_{\text{out}} = -9.05 \times 10^{-7}\Lambda^2 + 1.08 \times 10^{-4}\Lambda - 0.080$ |

The simplified form of the longitudinal small disturbance linearization model can be expressed as follows:

$$\begin{bmatrix} \Delta\dot{V} \\ \Delta\dot{\alpha} \\ \Delta\dot{q} \\ \Delta\dot{\theta} \end{bmatrix} = A \begin{bmatrix} \Delta V \\ \Delta\alpha \\ \Delta q \\ \Delta\theta \end{bmatrix} + B \begin{bmatrix} \Delta\delta_e \\ \Delta\Lambda \end{bmatrix} \tag{11}$$

where the input matrix includes the equivalent elevator deflection of the V-tail and the increase in the aerodynamic force, caused by the coordinated deformation of the inner and outer wing sections.

In this study, the lateral/directional small-disturbance linearization model is constructed for the special case of the deflection of the trailing edge of the left outer section and the change in the sweep angle of the right outer section. In addition, the influence of the roll damping is considered. The simplified form of the linearization model is expressed as follows:

$$\begin{bmatrix} \Delta\dot{\beta} \\ \Delta\dot{p} \\ \Delta\dot{r} \\ \Delta\dot{\phi} \end{bmatrix} = C \begin{bmatrix} \Delta\beta \\ \Delta p \\ \Delta r \\ \Delta\phi \end{bmatrix} + D \begin{bmatrix} \Delta\delta_r \\ \Delta\delta_{a,2} \\ \Delta\Lambda_4 \end{bmatrix} \tag{12}$$

where $\triangle\delta_{a,2}$ represents the deflection angles at the trailing edge of the left outer section, which affects the aerodynamic forces of the left inner and outer sections; and $\triangle\Lambda_4$ represents the sweep angle change in the right outer section, which affects the aerodynamic forces of the right inner and outer sections.

### 5.3. Influence Analysis of Dynamic Stability

According to the configuration and flight conditions of the multi-section variable-sweep wing UAV studied in this research, the flight quality of the targeted UAV can be preliminarily evaluated according to the requirements of type I aircraft in flight phase B [47].

For the longitudinal pitch motion, the long-period modal of the UAV reaches level 1 quality, but the damping ratio of the short-period modal is insufficient. The CAP range for level 1 flight quality requirements is 0.085~0.36, and the damping ratio is required to be 0.3~2. Therefore, it is necessary to introduce the angle of attack and the pitch rate for feedback compensation. For the lateral/directional motion, the roll modal achieves

level 1 quality, but the Dutch roll modal does not meet the requirements for level 1 quality ($\xi_{RS}\omega_{RS} > 0.5$ rad/s), and feedback compensation is also required.

In this section, stability augmentation adjustment is performed for the longitudinal and lateral/directional motions separately.

### 5.3.1. Longitudinal Stability Augmentation Design

The augmentation model shown in Figure 5 is adopted in this research. For convenience, the model is adjusted with an angle of attack filter and without a pitch rate filter. For the angle of attack feedback, the dynamic characteristics of the actuator and the filter are expressed as first-order inertia elements, with bandwidths of 10 rad/s and 20.2 rad/s, respectively [48], as follows:

$$G_\alpha(s) = \frac{10}{s+10}, \; G_\delta(s) = \frac{20.2}{s+20.2} \tag{13}$$

The angle of attack filter can be taken as a part of the dynamics model, thus the state variables of the system are $x = [V,\alpha,q,\theta,x_\alpha,x_F]$. Because the positive elevator deflection generates a negative pitching moment, an inverter is added between the elevator actuator and the control surface, to adopt the positive gain root locus method for design. Based on the above analysis, the augmented dynamic equation of the system can be expressed as follows:

$$\dot{x} = \begin{bmatrix} & & & & 0 \\ A & & -B & & 0 \\ & & & & 0 \\ & & & & 0 \\ 0 & 0 & 0 & 0 & -20.2 & 0 \\ 0 & 10 & 0 & 0 & 0 & -10 \end{bmatrix} \begin{bmatrix} V \\ \alpha \\ q \\ \theta \\ x_\alpha \\ x_F \end{bmatrix} + \begin{bmatrix} 0 \\ 0 \\ 0 \\ 0 \\ 20.2 \\ 0 \end{bmatrix} u_e, \; y = \begin{bmatrix} & & C & & \\ 0 & 0 & 0 & 0 & 0 & 1 \end{bmatrix} x \tag{14}$$

The angle of attack feedback gain $k_\alpha = 0.3$ and pitch angle velocity feedback gain $k_q = 0.25$ are introduced in this research to improve the flight quality of the short-period modal. With the same parameters of the stability augmentation system, the pre-selected models are adjusted under different conditions. Figure 12 shows the changes in the short-period characteristic roots, with the configuration after stability augmentation under different Mach numbers, where the arrow indicates that the sweep angle of the outer section $\Lambda$out, corresponding to the configuration, decreases linearly in the coordinated deformation process of the wing.

When the flight speed is small, the characteristic root of the system gradually moves away from the imaginary axis. The system stability is enhanced with a decrease in $\Lambda_{out}$, corresponding to the configurations. For the model with a relatively small wingspan proportion of the inner section, the short-period modal presents a change law that first approaches the real axis and then moves away from the real axis, with a decrease in $\Lambda_{out}$, corresponding to the configurations. For the model with a large wingspan proportion of the inner section, the short-period mode gradually approaches the real axis.

The trim angle of attack is negative and the short-period modal gradually moves away from the real axis when the flight speed is high. With an increase in the wingspan proportion of the inner section of the pre-selected models, the variation range of the characteristic roots decreases gradually, and the influence of the configuration changes on the short-period modes decreases gradually.

Figures 13 and 14 show the short-period damping ratio $\xi$ and oscillation frequency $\omega_n$ of the UAV system after stability augmentation adjustment. Long-period modal characteristics are less affected. The following conclusions can be drawn:

1.  With the increase in the wingspan proportion of the inner section of the pre-selected models, the effect of the longitudinal stability augmentation system on the longitudinal short-period modal of the targeted UAV is gradually weakened. The short-period damping ratio $\xi$ decreases with an increase in the wingspan proportion of the inner

section, after the stability augmentation adjustment. The difference between the short-period damping ratio $\zeta$ of the pre-selected model increases with the sweep angle of the outer section $\Lambda_{out}$, corresponding to the configuration, and the $C_{Dmin}$ configuration, corresponding to the minimum damping ratio $\zeta$ of the pre-selected models.

2. In the case where the trim angle of attack is positive (the curves corresponding to Ma = 0.3 in Figure 13), with the increase in the wingspan proportion of the inner section, the oscillation frequency $\omega_n$ gradually transitions from monotone decreasing to monotone increasing, with the increase in the sweep angle of the outer section. Among them, the variation range of the oscillation frequency $\omega_n$ of Model3 and Model4 is small during the entire deformation process, and the value of $\omega_n$ is large in different configurations. At this time, the wingspan ratio of the inner and outer sections ranges from 1.41 to 1.78.

3. When the trim angle of attack is negative (the curves corresponding to Ma = 0.5, as shown in Figure 14), the change in wing configurations has little effect on $\zeta$ and $\omega_n$ for the models with a large wingspan proportion of the inner section. For the models with a relatively small wingspan proportion of the inner section, the value of $\zeta$ increases gradually and $\omega_n$ decreases gradually with an increase in $\Lambda_{out}$, corresponding to the configurations.

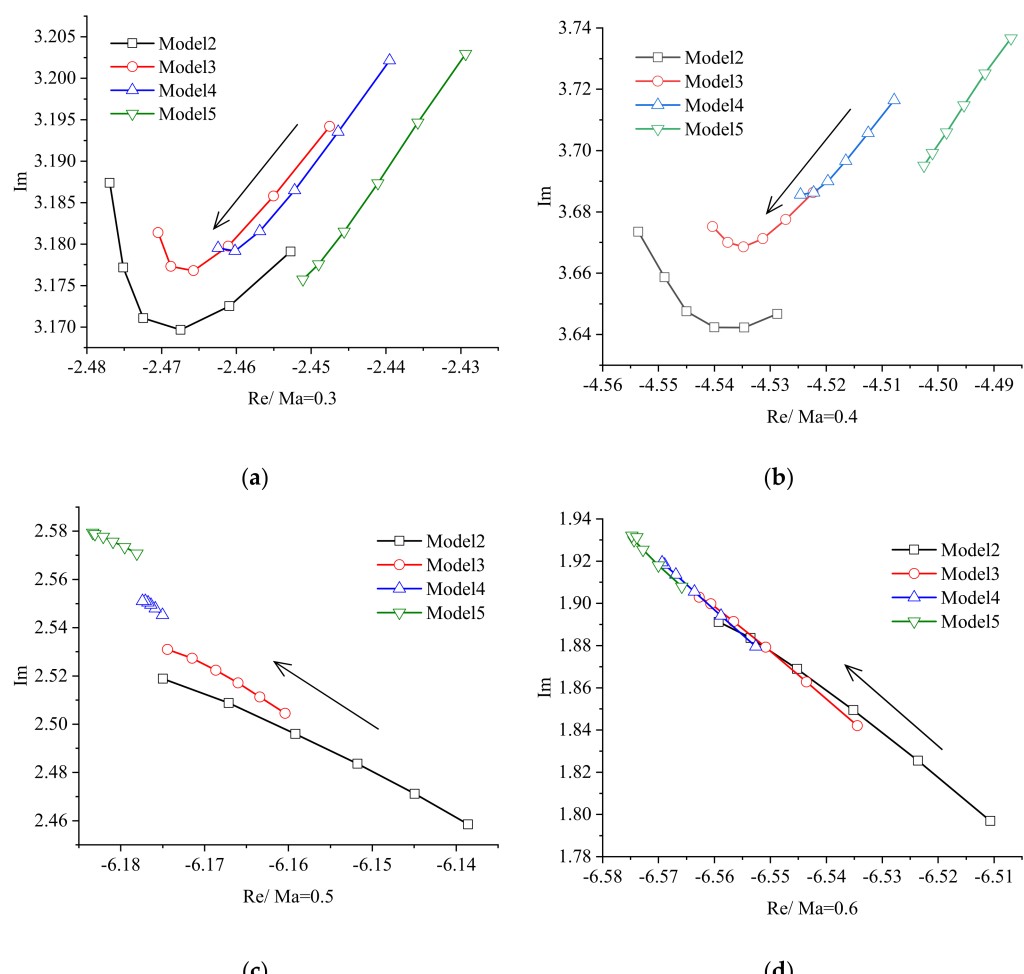

**Figure 12.** The changes in the short-period characteristic roots of the system with the configuration after stabilization augmentation under the following different Mach numbers: (**a**) Ma = 0.3, (**b**) Ma = 0.4, (**c**) Ma = 0.5, (**d**) Ma = 0.6.

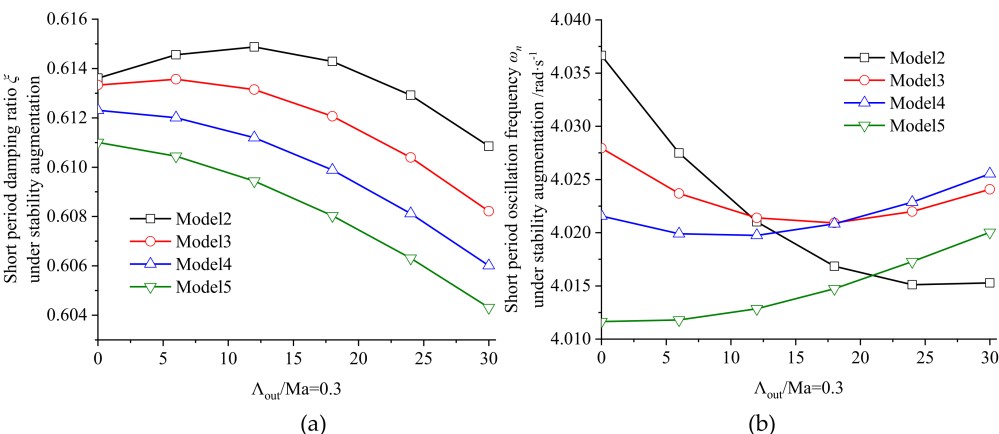

**Figure 13.** The short-period damping ratio $\xi$ and oscillation frequency $\omega_n$ of the UAV system after stability augmentation adjustment when Ma = 0.3. (**a**) Damping ratio $\xi$, (**b**) oscillation frequency $\omega_n$.

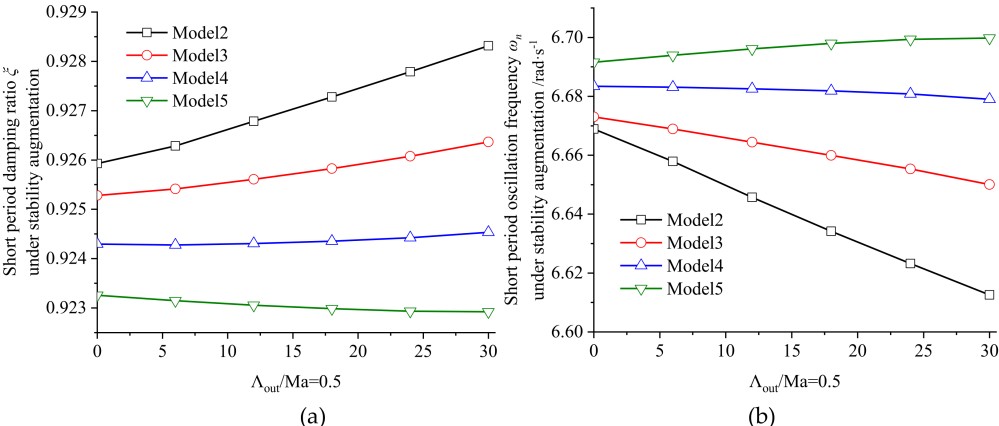

**Figure 14.** The short-period damping ratio $\xi$ and oscillation frequency $\omega_n$ of the UAV system after stability augmentation adjustment when Ma = 0.5. (**a**) Damping ratio $\xi$, (**b**) oscillation frequency $\omega_n$.

### 5.3.2. Lateral/Directional Stability Augmentation Design

The aileron actuator, sweep angle actuator, and rudder actuator are all regarded as simple hysteresis elements, and the transition frequency is 30. The bending modal filter is ignored. Because the positive control surface deflection produces a negative torque, an inverter is added to the output matrix of the control surface actuator. At the same time, considering that the asymmetric change in sweep angle can be used as an additional input for the lateral/directional maneuvering control, the rudder, aileron, and sweep angle are combined into a three-input and two-output system, which is connected in series with the controlled system.

$$\boldsymbol{a}_a = \begin{bmatrix} -30 & 0 & 0 \\ 0 & -30 & 0 \\ 0 & 0 & -30 \end{bmatrix}, \boldsymbol{b}_a = \begin{bmatrix} 30 & 0 & 0 \\ 0 & 30 & 0 \\ 0 & 0 & 30 \end{bmatrix}, \boldsymbol{c}_a = \begin{bmatrix} -1 & 0 & 0 \\ 0 & -1 & 0 \\ 0 & 0 & -1 \end{bmatrix}, \boldsymbol{d}_a = \begin{bmatrix} 0 & 0 & 0 \\ 0 & 0 & 0 \\ 0 & 0 & 0 \end{bmatrix} \tag{15}$$

The washout filter is combined into the three-input and four-output models, which are directly connected to the first pair of inputs and outputs, which are as follows:

$$\boldsymbol{a}_w = [-1/\tau_w], \boldsymbol{b}_w = \begin{bmatrix} 0 & 1/\tau_w \end{bmatrix}, \boldsymbol{c}_w = \begin{bmatrix} 0 \\ -1 \end{bmatrix}, \boldsymbol{d}_w = \begin{bmatrix} 1 & 0 \\ 0 & 1 \end{bmatrix} \tag{16}$$

where the time constant of the external washing filter is a compromised value. If the time constant is too large, the yaw damper impedes turn entry [38]. In this research, the time

constant $\tau_w$ = 1.05 s, and the closed-loop feedback gains of the roll rate and yaw rate loop are $K_p$ = 0.2 and $K_r$ = 1.6, respectively.

Figures 15 and 16 show the damping ratio $\zeta$ and the oscillation frequency $\omega_n$ of the Dutch roll modal of the pre-selected model, after the stability augmentation adjustment, with $\Lambda_{out}$ corresponding to the configurations at different Mach numbers. In general, the value of $\zeta$ increases first and then decreases with an increase in $\Lambda_{out}$.

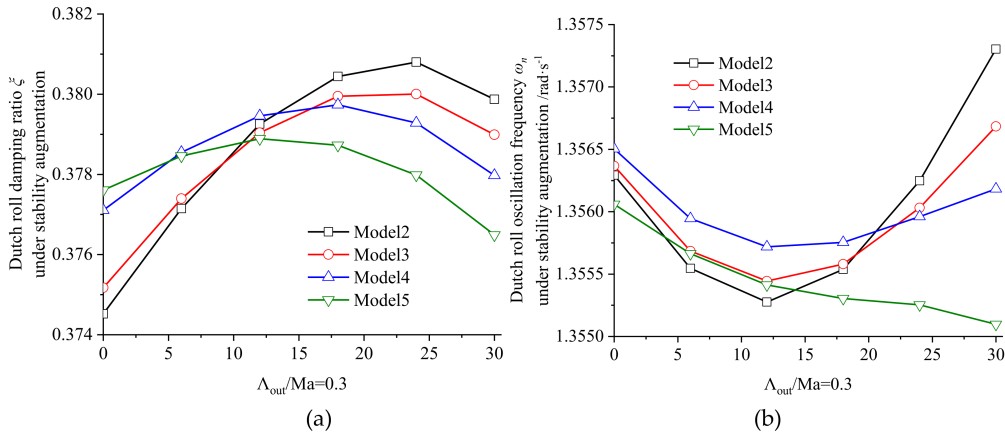

(a)                                                    (b)

**Figure 15.** The damping ratio $\zeta$ and the oscillation frequency $\omega_n$ of the Dutch roll modal of the pre-selected model after the stability augmentation adjustment with $\Lambda$out corresponding to the configurations when Ma = 0.3. (**a**) Damping ratio $\zeta$, (**b**) oscillation frequency $\omega_n$.

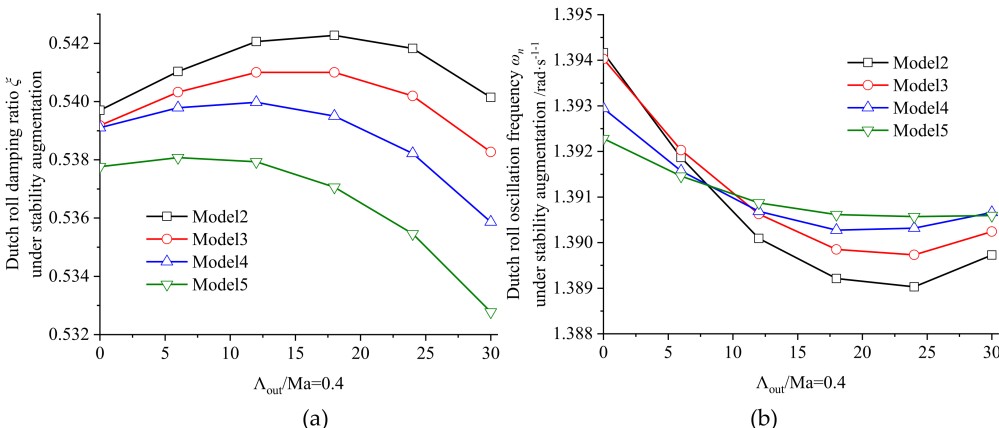

(a)                                                    (b)

**Figure 16.** The damping ratio $\zeta$ and the oscillation frequency $\omega_n$ of the Dutch roll modal of the pre-selected models after the stability augmentation adjustment with $\Lambda$out corresponding to the configurations when Ma = 0.4. (**a**) Damping ratio $\zeta$, (**b**) oscillation frequency $\omega_n$.

When the flight speed is low and $\Lambda_{out}$, corresponding to the configurations, is small, the Dutch roll damping ratio $\zeta$ increases, and the oscillation frequency $\omega_n$ increases first and then decreases monotonically with the wingspan proportion of the inner section. When $\Lambda_{out}$, corresponding to the configurations, is large, the wingspan proportion of the inner section is greater, and the Dutch roll $\zeta$ and $\omega_n$ are smaller.

At higher flight speeds, the Dutch roll $\zeta$ decreases with an increase in the wingspan proportion of the inner section. When the wingspan proportion of the inner section is relatively large, and the $\Lambda_{out}$ corresponding to the configurations is relatively small, the configuration change has little effect on $\zeta$ of the Dutch roll modal. When the $\Lambda_{out}$ corresponding to the configurations is small, $\omega_n$ shows a monotonically changing rule of first increasing and then decreasing with the wingspan proportion of the inner section. When $\Lambda_{out}$, corresponding to the configurations, is large, $\omega_n$ gradually increases with the wingspan proportion of the inner section.

Tables 8 and 9 present the roll damping of pre-selected models under the stability augmentation adjustment when Ma = 0.3 and Ma = 0.4. Based on the calculation results of the lateral/directional modals of the multi-section variable-sweep wing UAV under the action of stability augmentation adjustment, it can be observed that with the increase in the wingspan proportion of the inner section, the adjustment effect of the stability augmentation system on the lateral/directional Dutch roll modal of the UAV is gradually weakened. With the increase in flight speed, the adjustment of the lateral/directional stability augmentation system is gradually enhanced. The following conclusions are drawn:

1. When the $\Lambda_{out}$, corresponding to the configurations, varies in the range of 0~15°, and the flight speed is low, Model4 has a larger value of Dutch roll $\xi$ and $\omega_n$. The multi-section variable-sweep wing UAV system exhibits better lateral/directional stability. At this time, the wingspan ratio of the inner and outer sections is approximately 1.78. At higher flight speeds, Model2 has better lateral/directional stability, followed by Model3, and the corresponding wingspan ratios of the inner and outer sections are 1.13 and 1.41, respectively.

2. When $\Lambda_{out}$, corresponding to the configurations, varies in the range of 15~30°, the Dutch roll $\xi$ decreases with an increase in the wingspan proportion of the inner section. When the flight speed is low, Model2 has better lateral/directional stability, followed by Model3, and the corresponding wingspan ratios of the inner and outer wing sections equal 1.13 and 1.41, respectively. At higher flight speeds, the oscillation period of Molde5 is shorter, followed by Model4, and the corresponding wingspan ratios of the inner and outer sections equal 2.28 and 1.78, respectively.

3. In addition, the change in configuration, and the wingspan ratio of the inner and outer wing sections of the pre-selected models have little effect on the rolling damping $\xi$ under the adjustment of the stability augmentation system. With the increase in flight speed, the absolute value of the roll damping $\xi$ increases gradually.

**Table 8.** The roll damping of pre-selected models under the stability augmentation adjustment when Ma = 0.3.

| $\Lambda_{out}/°$ | Model2 | Model3 | Model4 | Model5 |
|---|---|---|---|---|
| 30 | −2.13 | −2.14 | −2.15 | −2.16 |
| 24 | −2.13 | −2.13 | −2.14 | −2.15 |
| 18 | −2.12 | −2.13 | −2.14 | −2.14 |
| 12 | −2.11 | −2.12 | −2.13 | −2.14 |
| 6 | −2.10 | −2.11 | −2.12 | −2.13 |
| 0 | −2.09 | −2.10 | −2.12 | −2.13 |

**Table 9.** The roll damping of pre-selected models under the stability augmentation adjustment when Ma = 0.4.

| $\Lambda_{out}/°$ | Model2 | Model3 | Model4 | Model5 |
|---|---|---|---|---|
| 30 | −4.18 | −4.17 | −4.11 | −4.01 |
| 24 | −4.17 | −4.17 | −4.17 | −4.05 |
| 18 | −4.15 | −4.16 | −4.17 | −4.15 |
| 12 | −4.12 | −4.13 | −4.16 | −4.16 |
| 6 | −4.09 | −4.11 | −4.14 | −4.15 |
| 0 | −4.05 | −4.08 | −4.12 | −4.14 |

## 6. Discussions

This research verifies the advantages of the multi-section variable-sweep wing in balancing the contradiction between performance improvement and stability in the high-aspect-ratio heavy-duty UAV. The influences of configuration change, and the ratio of the wingspan length of the inner and outer wing sections on the performance of morphing UAV are investigated. It provides a reference for the practical application of the multi-section

variable-sweep wing technology in the high-aspect-ratio heavy-duty UAV. Simulation analysis verifies the effectiveness of the dynamic stability augmentation system. These results are all extremely encouraging and motivate further investigation of multi-section wing morphing technology. The following aspects should be covered in future research.

First, five pre-selected models are constructed, based on the distribution characteristics of wing structure in this research. The weight gain of the actuator and deformation structure adopts the assumption of uniform distribution, which is acceptable in the initial design stage of morphing UAVs. In fact, the weight gain distribution of the actuator and deformation structure is not a negligible quantity, and the effect on aircraft performance may be decisive in certain flight environments. This research takes the pre-selected models with different mass ratios of wing sections as the research object, and shows the influence of actuator and structural weight distribution on aircraft performance in a broader scope. In the follow-up research work, the corresponding structural weight distribution characteristics can be introduced into the centroid offset correction function, and the results can be further optimized.

Second, the optimal configuration of the morphing UAV may undergo major changes when performing different tasks, which, in turn, cause major changes in the moment of inertia, aerodynamic characteristics, and stability. Therefore, it is necessary to design the flight control system for different tasks, to ensure flight stability during the deformation process, and to explore the influence of configuration changes and the proportion of the wingspan length of the inner and outer wing sections on the flight control. In addition, the asymmetric sweep angle change provides a new control mode for lateral maneuvers. Specific control algorithms are required to explore the application conditions of asymmetric morphing in maneuver flights and improve the control efficiency of UAVs under different conditions.

Third, the deformation mechanism moves under the action of the internal actuators, to realize the configuration change in the UAV, or the deflection of the control surface. The aerodynamic load distribution on the wing surface directly affects the actuators' design and efficiency of the control surface. Therefore, how to quickly obtain the pressure distribution on the wing surface, and obtain the range of the wingspan ratio of the inner and outer sections with low performance requirements for actuators have become the important problems to be solved in the overall design stage of multi-section wing morphing UAV.

## 7. Conclusions

In this study, a multi-section variable-sweep wing unmanned aerial vehicle (UAV) design scheme is proposed, which realizes the centroid self-trim compensation through the reverse cooperative sweep angle change in the inner and outer wing sections. While improving the aerodynamic performance, stability, and maneuvering, reducing the influence of the movement of the centroid and aerodynamic center and the change in the moment of inertia of the high-aspect-ratio morphing UAV. Based on the overall design results, and considering the centroid movement caused by the wing structure deformation, the performance evaluation of the targeted bionic morphing UAV under different working conditions is carried out from the following two aspects: aerodynamic characteristics analysis, and stability analysis and augmentation adjustment. The influences of the Mach number, angle of attack, sweep angle, and wingspan ratio of the inner and outer sections are discussed in detail.

Based on the results of the calculation and analysis, the following main conclusions can be drawn:

1.  The influence of sweep angle change on the longitudinal static stability margin of the high-aspect-ratio UAV can be greatly improved by the coordinate deformation mode of a multi-section variable-sweep wing. Reducing the mass of components near the centroid, for example, with an increase in fuel consumption, the static stability margin of the multi-section variable-sweep wing UAV designed in this research is gradually increased.

2. The analysis results show that when the sweep angle of the outer section, corresponding to the configuration, is in the range of $0°\sim10°$, the aerodynamic force of the UAV changes a little. Among them, the slope difference of the lift line, corresponding to Model3 and Model5, is small. The lift/drag ratios, corresponding to Model4 and Model5, at different angles of attack have a small difference.

3. The longitudinal pitch trimming results of different configurations of the pre-selected models, under steady and flat flight conditions, show that properly increasing the wingspan proportion of the inner section can reduce the trim resistance of tail and engine thrust, and, at the same time, increase the range and duration.

4. The configuration change has little effect on the long-period modal and roll damping under a given working condition. Under the adjustment of the stability augmentation system, the configuration with a smaller sweep angle of the outer section ($\Lambda_{out}$ = $0\sim10°$) has better longitudinal stability. Increasing the sweep angle of the outer section, corresponding to the configuration, can improve the lateral/directional stability of the UAV. Considering the influence of the wingspan ratio of the inner and outer sections on the efficiency of the control surface, the pre-selected model of Model3 to Model4 has better dynamic stability characteristics. The damping ratio and oscillation frequency are all in the optimal or suboptimal state, and the range of the value change is small.

Based on the analysis results of the aerodynamic characteristics, dynamic stability, and augmentation adjustment of the pre-selected models under the centroid self-trim compensation morphing mode, Model3 has better longitudinal stability and Model4 has better lateral/directional stability. The corresponding wingspan ratio of the inner and outer sections ranges from 1.41 to 1.78.

**Author Contributions:** All the authors conceived the idea and developed the method. H.M. contributed to the formulation of methodology and original draft. Y.G. contributed to the data curation and supervision. B.S. and Y.P. contributed to the editing. All authors have read and agreed to the published version of the manuscript.

**Funding:** This study was supported by Research Funds for Interdisciplinary Subject of Northwestern Polytechnical University (NWPU). No. 19SH0304017.

**Institutional Review Board Statement:** Not applicable.

**Informed Consent Statement:** Not applicable.

**Conflicts of Interest:** The authors declare no conflict of interest.

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
