# Peer review of "Stability Analysis and Augmentation Design of a Bionic Multi-Section Variable-Sweep-Wing UAV Based on the Centroid Self-Trim Compensation Morphing"

_applsci, doi:10.3390/app11198859_

Round 1
Reviewer 1 Report
The presented article is of a scientific and research nature.
Suggestions:
Please follow the manuscript preparation instructions, see figures 1, 2, 3, 4, 5, 6, 9, 13, center
- Expand the UAV (Unmanned Aerial Vehicle) abbreviation, see Summary, Keywords and Introduction
- Correct the descriptions to figures 7, 8, 10, 12, markings (a), (b) ....
- There is no figure 17 and 18 !!! see pages 20/651, 659
- Correct bibliographies, see pages 2/61, 2/84, page 18/567, page 21/684, organize the bibliographies
- What new and important aspects the work brings to learning - short description

Author Response
Point 1: Please follow the manuscript preparation instructions, see figures 1, 2, 3, 4, 5, 6, 9, 13, center.
Response 1: Thanks to the expert for your suggestions. Figures 1, 2, 3, 4, 5, 6, 11 have been centered in the article. The readability of the changed article has been improved.
Point 2: Expand the UAV (Unmanned Aerial Vehicle) abbreviation, see Summary, Keywords and Introduction
Response 2: Thanks to the expert for your advices. The UAV abbreviation has been change to Unmanned Aerial Vehicle where it first appeared in the abstract, introduction and summary.
Point 3: Correct the descriptions to figures 7, 8, 10, 12, markings (a), (b)
Response 3: Thanks to the expert for your advices. We have corrected the descriptions to figures 7, 8, 10, 12, 13, 14, 15 and 16, markings (a), (b)…
Point 4: There is no figure 17 and 18 !!! see pages 20/651, 659.
Response 4: Thanks to the expert for your advices. Due to the negligence of the authors, there are no figure 17 and 18 in the article. We have corrected Figure.17 in page 20/650 to Figure.13 and Figure.18 in page 20/658 to Figure.14.
Point 5: Correct bibliographies, see pages 2/61, 2/84, page 18/567, page 21/684, organize the bibliographies
Response 5: Thanks to the expert for your advices. Due to the negligence of the authors, there are indeed some problems in the citation of the bibliographies. We have changed 2/61 to 2/24, 2/84 to 2/34, 18/158 to 18/38, and checked the order of citation of bibliographies.
Point 6: What new and important aspects the work brings to learning - short description
Response 6: Thanks to the expert for your advices. In order to supplement the significance and innovation of the research as well as the future research prospects, the article added the Discussions section and modified the
Conclusions. 6. Discussions This research verifies the advantages of multi-section variable-sweep-wing in balancing the contradiction between performance improvement and stability in the high aspect ratio combat UAV. The influences of configuration change and the ratio of wingspan length of inner and outer wing sections on the performance of morphing UAV are investigated. It provides a reference for the practical application of the multi-section variable-sweep wing technology in the high aspect ratio combat UAV. Simulation analysis verifies the effectiveness of the dynamic stability augmentation system. These results are all extremely encouraging and motivate further investigation of multi-section wing morphing technology. The following aspects should be covered in future research. First, five pre-selected models are constructed based on the distribution characteristics of wing structure in this research. The weight gain of actuators and deformation structure adopts the assumption of uniform distribution, which is acceptable in the initial design stage of morphing UAVs. In fact, the weight gain distribution of actuator and deformation structure is not a negligible quantity, and the effect on aircraft performance may be decisive in certain flight environments. This research takes the pre-selected models with different mass ratio of wing sections as the research object, and shows the influence of actuator and structural weight distribution on aircraft performance in a broader scope. In the follow-up research work, the corresponding structural weight distribution characteristics can be introduced into the centroid offset correction function, and the results can be further optimized. Second, the optimal configuration of the morphing UAV may undergo major changes when performing different tasks, which in turn cause major changes in the moment of inertia, aerodynamic characteristics, and stability. Therefore, it is necessary to design the flight control system for different tasks to ensure flight stability during the deformation process and to explore the influence of configuration changes and the proportion of the wingspan length of the inner and outer wing sections on the flight control. In addition, the asymmetric sweep angle change provides a new control mode for lateral maneuver. Specific control algorithms are required to explore the application conditions of asymmetric morphing in maneuver flight and improve the control efficiency of UAVs under different conditions. Third, the deformation mechanism moves under the action of the internal actuators to realize the configuration change of the UAV or the deflection of the control surface. The aerodynamic load distribution on the wing surface directly affects the actuators’ design and efficiency of the control surface. Therefore, how to quickly obtain the pressure distribution on the wing surface and obtain the range of the wingspan ratio of the inner and outer sections with low performance requirements for actuators have become the important problems to be solved in the overall design stage of multi-section wing morphing UAV. 7. Conclusions In this study, a multi-section variable-sweep-wing UAV design scheme is proposed, which realizes the centroid self-trim compensation through the reverse cooperative sweep angle change of the inner and outer wing sections. While improving the aerodynamic performance, stability, and maneuvering, reducing the influence of the movement of the centroid and aerodynamic center and the change in the moment of inertia of the high-aspect-ratio morphing UAV. Based on the overall design results and considering the centroid movement caused by the wing structure deformation, the performance evaluation of the targeted bionic morphing UAV under different working conditions is carried out from two aspects: aerodynamic characteristics analysis, stability analysis and augmentation adjustment. The influences of the Mach number, angle of attack, sweep angle, and wingspan ratio of the inner and outer sections are discussed in detail. Based on the results of the calculation and analysis, the following main conclusions can be drawn: (1) The influence of sweep angle change on the longitudinal static stability margin of the high-aspect-ratio UAV can be greatly improved by the coordinate deformation mode of a multi-section variable-sweep-wing. Reducing the mass of components near the centroid, for example, with an increase in fuel consumption, the static stability margin of the multi-section variable-sweep-wing UAV designed in this research is gradually increased. (2) The analysis results show that when the sweep angle of the outer section corresponding to the configuration is in the range of 0°~10°, the aerodynamic force of the UAV changes little. Among them, the slope difference of the lift line corresponding to Model3 and Model5 is small. The lift-drag ratios corresponding to Model4 and Model5 at different angles of attack have a small difference. (3) The longitudinal pitch trimming results of different configurations of the pre-selected models under steady and flat flight conditions show that properly increasing the wingspan proportion of the inner section can reduce the trim resistance of tail and engine thrust, and at the same time increase the range and duration. (4) The configuration change has little effect on the long-period modal and roll damping under a given working condition. Under the adjustment of the stability augmentation system, the configuration with a smaller sweep angle of the outer section (Λout=0~10°) has better longitudinal stability. Increasing the sweep angle of the outer section corresponding to the configuration can improve the lateral-directional stability of the UAV. Considering the influence of the wingspan ratio of the inner and outer sections on the efficiency of the control surface, the pre-selected model of Model3 to Model4 has better dynamic stability characteristics. The damping ratio and oscillation frequency are all in the optimal or suboptimal state, and the range of the value change is small. Based on the analysis results of the aerodynamic characteristics, dynamic stability and augmentation adjustment of the pre-selected models under the centroid self-trim compensation morphing mode, Model3 has better longitudinal stability and Model4 has better lateral/directional stability. The corresponding wingspan ratio of the inner and outer sections ranges from 1.41 and 1.78.

Reviewer 2 Report
The work is interesting and provide a good background. The results show some improvements as a result of the proposed modifications.
I would recomend to provide a discussion section with a resume of the results and how they impove the state of the art. Also, some future directions could be provided considering the new achievements.
Author Response
Point 1: I would recomend to provide a discussion section with a resume of the results and how they impove the state of the art. Also, some future directions could be provided considering the new achievements.
Response 1: Thanks to the expert for your suggestions. In order to supplement the significance and innovation of the research as well as the future research prospects, the article added the Discussions section and modified the Conclusions.
- Discussions
This research verifies the advantages of multi-section variable-sweep-wing in balancing the contradiction between performance improvement and stability in the high aspect ratio combat UAV. The influences of configuration change and the ratio of wingspan length of inner and outer wing sections on the performance of morphing UAV are investigated. It provides a reference for the practical application of the multi-section variable-sweep wing technology in the high aspect ratio combat UAV. Simulation analysis verifies the effectiveness of the dynamic stability augmentation system. These results are all extremely encouraging and motivate further investigation of multi-section wing morphing technology. The following aspects should be covered in future research.
First, five pre-selected models are constructed based on the distribution characteristics of wing structure in this research. The weight gain of actuators and deformation structure adopts the assumption of uniform distribution, which is acceptable in the initial design stage of morphing UAVs. In fact, the weight gain distribution of actuator and deformation structure is not a negligible quantity, and the effect on aircraft performance may be decisive in certain flight environments. This research takes the pre-selected models with different mass ratio of wing sections as the research object, and shows the influence of actuator and structural weight distribution on aircraft performance in a broader scope. In the follow-up research work, the corresponding structural weight distribution characteristics can be introduced into the centroid offset correction function, and the results can be further optimized.
Second, the optimal configuration of the morphing UAV may undergo major changes when performing different tasks, which in turn cause major changes in the moment of inertia, aerodynamic characteristics, and stability. Therefore, it is necessary to design the flight control system for different tasks to ensure flight stability during the deformation process and to explore the influence of configuration changes and the proportion of the wingspan length of the inner and outer wing sections on the flight control. In addition, the asymmetric sweep angle change provides a new control mode for lateral maneuver. Specific control algorithms are required to explore the application conditions of asymmetric morphing in maneuver flight and improve the control efficiency of UAVs under different conditions.
Third, the deformation mechanism moves under the action of the internal actuators to realize the configuration change of the UAV or the deflection of the control surface. The aerodynamic load distribution on the wing surface directly affects the actuators’ design and efficiency of the control surface. Therefore, how to quickly obtain the pressure distribution on the wing surface and obtain the range of the wingspan ratio of the inner and outer sections with low performance requirements for actuators have become the important problems to be solved in the overall design stage of multi-section wing morphing UAV.
- Conclusions
In this study, a multi-section variable-sweep-wing UAV design scheme is proposed, which realizes the centroid self-trim compensation through the reverse cooperative sweep angle change of the inner and outer wing sections. While improving the aerodynamic performance, stability, and maneuvering, reducing the influence of the movement of the centroid and aerodynamic center and the change in the moment of inertia of the high-aspect-ratio morphing UAV. Based on the overall design results and considering the centroid movement caused by the wing structure deformation, the performance evaluation of the targeted bionic morphing UAV under different working conditions is carried out from two aspects: aerodynamic characteristics analysis, stability analysis and augmentation adjustment. The influences of the Mach number, angle of attack, sweep angle, and wingspan ratio of the inner and outer sections are discussed in detail.
Based on the results of the calculation and analysis, the following main conclusions can be drawn:
1)The influence of sweep angle change on the longitudinal static stability margin of the high-aspect-ratio UAV can be greatly improved by the coordinate deformation mode of a multi-section variable-sweep-wing. Reducing the mass of components near the centroid, for example, with an increase in fuel consumption, the static stability margin of the multi-section variable-sweep-wing UAV designed in this research is gradually increased.
2)The analysis results show that when the sweep angle of the outer section corresponding to the configuration is in the range of 0°~10°, the aerodynamic force of the UAV changes little. Among them, the slope difference of the lift line corresponding to Model3 and Model5 is small. The lift-drag ratios corresponding to Model4 and Model5 at different angles of attack have a small difference.
3)The longitudinal pitch trimming results of different configurations of the pre-selected models under steady and flat flight conditions show that properly increasing the wingspan proportion of the inner section can reduce the trim resistance of tail and engine thrust, and at the same time increase the range and duration.
4)The configuration change has little effect on the long-period modal and roll damping under a given working condition. Under the adjustment of the stability augmentation system, the configuration with a smaller sweep angle of the outer section (Λout=0~10°) has better longitudinal stability. Increasing the sweep angle of the outer section corresponding to the configuration can improve the lateral-directional stability of the UAV. Considering the influence of the wingspan ratio of the inner and outer sections on the efficiency of the control surface, the pre-selected model of Model3 to Model4 has better dynamic stability characteristics. The damping ratio and oscillation frequency are all in the optimal or suboptimal state, and the range of the value change is small.
Based on the analysis results of the aerodynamic characteristics, dynamic stability and augmentation adjustment of the pre-selected models under the centroid self-trim compensation morphing mode, Model3 has better longitudinal stability and Model4 has better lateral/directional stability. The corresponding wingspan ratio of the inner and outer sections ranges from 1.41 and 1.78.

Reviewer 3 Report
Review of “Stability Analysis and …” by Hang Ma and others
This is a well-written paper that addresses the interesting question of how variable-sweep-wing designs, and especially two-jointed ones inspired by bird wings, can be applied to improve the importance of the high-aspect ratio UAV. The analysis is mostly static, and focused on the ability of the two-jointed design to change the airfoil shape without causing major offset in the UAV’s centroid.
I very much like the sections that enumerate the conclusions being drawn from particular analyses. They make the results that much more compelling.
It is understandable that a military UAV (General Atomics MQ-9 Reaper) is chosen as the base airframe upon which design changes are proposed, because its is a large UAV in common use. However, since the results of the paper have application to both civilian/industrial and military UAV’s, my suggestion is that the warfare-specific words be replaced with more generally functional ones. Thus, for instance, the word “releasable-payload” can replace “bomb”, “heavy-duty” replace “combat”, etc. (Concerning payloads, I note that UAV’s are finding increased use for the delivery of medical supplies).
Although most of the figures are well-done and well-explained, Fig 4 remains a mystery to me, because the wing appears to lack any obvious “sections”, at least as they are depicted in Figure 3. I think this figure can be omitted.
By far, the weakest part of the paper is Section 4.1 on aerodynamic characteristics, because I do not get a clear sense for how quantities (such as drag coefficients) are being calculated. This may be explained to some degree in subsequent sections, but some of the material needs to be moved up.
On line 508, “O sub f” is mistakenly rendered “Of”.
Not much is said about the implications of different rates of wing-section movement. I would have thought that would be an important aspect of the design.
Overall, it’s a good paper and can be published with minor revision.

Author Response
Point 1: It is understandable that a military UAV (General Atomics MQ-9 Reaper) is chosen as the base airframe upon which design changes are proposed, because its is a large UAV in common use. However, since the results of the paper have application to both civilian/industrial and military UAV’s, my suggestion is that the warfare-specific words be replaced with more generally functional ones. Thus, for instance, the word “releasable-payload” can replace “bomb”, “heavy-duty” replace “combat”, etc. (Concerning payloads, I note that UAV’s are finding increased use for the delivery of medical supplies).
Response 1: Thanks to the expert for your suggestions. As you said, the results of the paper have application to both civilian/industrial and military UAVs. Therefore, we replace the word “bomb” to “releasable-payload” (5/208, 10/311, 10/329, 10/311, 10/336, 11/348, 11/349) and “combat” to “heavy-duty” (2/97, 3/101, 3/120, 23/745, 23/748).
Point 2: Although most of the figures are well-done and well-explained, Fig 4 remains a mystery to me, because the wing appears to lack any obvious “sections”, at least as they are depicted in Figure 3. I think this figure can be omitted.
Response 2: Thanks to the expert for your advices. As you said, the wing structure in Figure.4 appears to lack any obvious sections. The distribution of wing stringers and ribs affects the subsequent coordinated deformation of inner and outer wing sections. Figure.4 can clearly show the design results of the wing structure, in which the inner and outer sections are presented as a whole part. We think the addition of Figure.4 makes the article more scientific and credible, so we still want to keep Figure.4 in this article. We hope you can understand our idea.
Point 3: By far, the weakest part of the paper is Section 4.1 on aerodynamic characteristics, because I do not get a clear sense for how quantities (such as drag coefficients) are being calculated. This may be explained to some degree in subsequent sections, but some of the material needs to be moved up.
Response 3: Thanks to the expert for your advices. Due to the negligence of the authors, the calculation method of aerodynamic characteristics does not illustrate in the manuscript. A supplementary explanation has been made in the revised version (11/359)。
Therefore, the aerodynamic analysis of the pre-selected models based on the coordinated deformation mode is performed using Fluent. The flow control equation is Reynolds average N-S equation, and the turbulence model adopts S-A (Spalart-Allmaras) model. Subsequently, a multivariable influence analysis is carried out on the longitudinal pitch trim characteristics.
Point 4: On line 508, “O sub f” is mistakenly rendered “Of”.
Response 4: Thanks to the expert for your advices. We have corrected “Of” to “Of” (15/507)
Point 5: Not much is said about the implications of different rates of wing-section movement. I would have thought that would be an important aspect of the design.
Response 5: Thanks to the expert for your suggestions. As you said, the change rates of wing sections an important aspect of the targeted UAV design. In this research, the performance evaluation of the targeted bionic morphing UAV under different working conditions is carried out from two aspects: aerodynamic characteristics analysis, stability analysis and augmentation adjustment. The quasi-steady assumption is used to analyze the performance under the small deformation rate of wing sections. As the focus of our next research, the influence of deformation rate on aircraft performance can be presented in detail through the control characteristics analysis of UAV under different working conditions. In order to supplement the significance and innovation of the research as well as the future research prospects, the article added the Discussions section.
- Discussions
This research verifies the advantages of multi-section variable-sweep-wing in balancing the contradiction between performance improvement and stability in the high aspect ratio combat UAV. The influences of configuration change and the ratio of wingspan length of inner and outer wing sections on the performance of morphing UAV are investigated. It provides a reference for the practical application of the multi-section variable-sweep wing technology in the high aspect ratio combat UAV. Simulation analysis verifies the effectiveness of the dynamic stability augmentation system. These results are all extremely encouraging and motivate further investigation of multi-section wing morphing technology. The following aspects should be covered in future research.
First, five pre-selected models are constructed based on the distribution characteristics of wing structure in this research. The weight gain of actuators and deformation structure adopts the assumption of uniform distribution, which is acceptable in the initial design stage of morphing UAVs. In fact, the weight gain distribution of actuator and deformation structure is not a negligible quantity, and the effect on aircraft performance may be decisive in certain flight environments. This research takes the pre-selected models with different mass ratio of wing sections as the research object, and shows the influence of actuator and structural weight distribution on aircraft performance in a broader scope. In the follow-up research work, the corresponding structural weight distribution characteristics can be introduced into the centroid offset correction function, and the results can be further optimized.
Second, the optimal configuration of the morphing UAV may undergo major changes when performing different tasks, which in turn cause major changes in the moment of inertia, aerodynamic characteristics, and stability. Therefore, it is necessary to design the flight control system for different tasks to ensure flight stability during the deformation process and to explore the influence of configuration changes and the proportion of the wingspan length of the inner and outer wing sections on the flight control. In addition, the asymmetric sweep angle change provides a new control mode for lateral maneuver. Specific control algorithms are required to explore the application conditions of asymmetric morphing in maneuver flight and improve the control efficiency of UAVs under different conditions.
Third, the deformation mechanism moves under the action of the internal actuators to realize the configuration change of the UAV or the deflection of the control surface. The aerodynamic load distribution on the wing surface directly affects the actuators’ design and efficiency of the control surface. Therefore, how to quickly obtain the pressure distribution on the wing surface and obtain the range of the wingspan ratio of the inner and outer sections with low performance requirements for actuators have become the important problems to be solved in the overall design stage of multi-section wing morphing UAV.

Round 2
Reviewer 2 Report
All issues were addressed.